# Aggregation of rhodopsin mutants in mouse models of autosomal dominant retinitis pigmentosa

Sreelakshmi Vasudevan ®[1], Subhadip Senapati ®[1,2], Maryanne Pendergast[1] & Paul S.–H. Park ®[1] ✉

Mutations in rhodopsin can cause it to misfold and lead to retinal degeneration. A distinguishing feature of these mutants in vitro is that they mislocalize and aggregate. It is unclear whether or not these features contribute to retinal degeneration observed in vivo. The effect of P23H and G188R misfolding mutations were examined in a heterologous expression system and knockin mouse models, including a mouse model generated here expressing the G188R rhodopsin mutant. In vitro characterizations demonstrate that both mutants aggregate, with the G188R mutant exhibiting a more severe aggregation profile compared to the P23H mutant. The potential for rhodopsin mutants to aggregate in vivo was assessed by PROTEOSTAT, a dye that labels aggregated proteins. Both mutants mislocalize in photoreceptor cells and PROTEOSTAT staining was detected surrounding the nuclei of photoreceptor cells. The G188R mutant promotes a more severe retinal degeneration phenotype and greater PROTEOSTAT staining compared to that promoted by the P23H mutant. Here, we show that the level of PROTEOSTAT positive cells mirrors the progression and level of photoreceptor cell death, which suggests a potential role for rhodopsin aggregation in retinal degeneration.

Retinitis pigmentosa (RP) is an inherited retinal degenerative disease that initially begins with the loss of rod photoreceptor cells, causing night blindness[1,2]. Rhodopsin is the light receptor in rod photoreceptor cells that initiates vision via phototransduction[3]. Over 100 mutations in rhodopsin have been detected in patients with congenital stationary night blindness or RP[4]. Rhodopsin mutations are the largest cause of autosomal dominant RP (adRP), which currently has no effective treatment or cure[1,4,5]. A majority of RP-causing mutations in rhodopsin with a known biochemical defect result in protein misfolding[4]. The severity of misfolding caused by mutation in rhodopsin is variable. Mutations can be broadly subdivided as either partial or complete misfolding mutations[6]. Based on in vitro biochemistry and cell biology characterizations, partial misfolding mutations result in mutants present as a variable mixture of folded and misfolded protein and can be chaperoned by retinoids, whereas complete misfolding mutations

result in mostly misfolded protein that cannot be chaperoned by retinoids[7–15].

Despite numerous in vitro and in vivo studies of misfolding mutants of rhodopsin since the initial discovery of the common P23H rhodopsin mutation[16], our understanding of the pathogenic mechanisms promoted by these mutations is still incomplete. The bulk of our understanding about misfolding rhodopsin mutants has come experimentally from in vitro cell culture systems and in vivo studies that have largely focused on animal models expressing the P23H rhodopsin mutant, although mice expressing other misfolding mutants have recently begun to emerge[17–19]. It is unclear in general what in vivo aspects the in vitro studies accurately model. The limited scope of in vivo studies examining, for the most part, a single misfolding mutation makes it difficult to assess misfolding mutants more broadly from different categories. Moreover, differences are observed

[1]Department of Ophthalmology and Visual Sciences, Case Western Reserve University, Cleveland, OH 44106, USA. [2]Present address: Prayoga Institute of Education Research, Bengaluru, KA 560116, India. ✉e-mail: paul.park@case.edu

depending on the type of animal model generated expressing the P23H rhodopsin mutant[20–25], presenting challenges in assessing pathogenic mechanisms.

While protein aggregates are a hallmark of many neurodegenerative diseases[26–29], it is unclear whether or not this is true in retinal degenerative diseases such as RP. Earlier in vitro studies on misfolding mutants of rhodopsin demonstrated that the mutants form aggregates that can be potentially toxic[30,31]. The aggregation of misfolding mutants of rhodopsin and their potential to harm photoreceptor cells in retinal degenerative diseases has yet to be established in vivo. In fact, rhodopsin mislocalization or aggregation was not detected in a knockin mouse model expressing the P23H rhodopsin mutant, previously generated and shown to best mimic the human phenotype compared to other transgenic animal models[20], questioning the validity of in vitro studies and whether rhodopsin aggregation plays a role in the pathogenesis of the disease.

In the current study, P23H and G188R rhodopsin mutations (Fig. 1A, B), both of which cause adRP[16,32], were examined in vitro and in vivo to determine the possibility that aggregation of the receptor contributes to retinal degeneration. The P23H mutation is a partial misfolding mutation that has been studied extensively both in vitro and in vivo whereas the G188R mutation is a complete misfolding mutation that has only been studied in vitro[13,33]. In vitro studies to characterize the aggregation properties of the two mutants were conducted in a heterologous expression system utilizing a Förster resonance energy transfer (FRET) based method[6]. In vivo studies were conducted in the previously characterized knockin mouse expressing the P23H rhodopsin mutant[20] and a mouse model generated in the current study where a G188R mutation was introduced into the rhodopsin gene by CRISPR/Cas9 technology (Fig. 1C, D). Studies here on the partial P23H misfolding mutation and the complete G188R

misfolding mutation in rhodopsin reveal how misfolding mutants from different subclasses manifest both in vitro and in vivo. These studies demonstrate the potential for rhodopsin aggregation to play a role in the retinal degeneration observed in mouse models of adRP.

## Results

### Aggregation and mislocalization of P23H and G188R rhodopsin in HEK293 cells

A distinguishing feature of misfolding rhodopsin mutants expressed in heterologous expression systems is that they are retained in the endoplasmic reticulum (ER) and form aggregates. We previously developed a FRET-based method to directly probe and quantify the aggregation of misfolding mutants of rhodopsin in transfected cells[6]. This method revealed variable aggregation properties for different misfolding mutants of rhodopsin including the P23H and G188R mutants on a human rhodopsin background[33]. Mutations on different animal backgrounds can result in different aggregation profiles[34]. Thus, aggregation profiles for the P23H and G188R mutations on a murine rhodopsin background were generated to help assess effects observed in mouse models presented later.

FRET curves were generated for mTq2- and YFP-tagged rhodopsin expressed in transfected HEK293 cells (Supplementary Fig. 1). FRET originating from rhodopsin oligomers and aggregates was differentiated by the ability of the mild detergent n-dodecyl-β-D-maltoside (DM) to disrupt the FRET signal. The total FRET signal is composed of DM-sensitive and DM-insensitive FRET, which originate from oligomers and aggregates of rhodopsin, respectively[6]. Properly folded rhodopsin forms oligomers whereas misfolded rhodopsin forms aggregates. The FRET signal must exceed the non-specific FRET, which we determined previously[33], to be considered specific FRET indicative of formation of physiologically relevant complexes. Wild-type (WT)

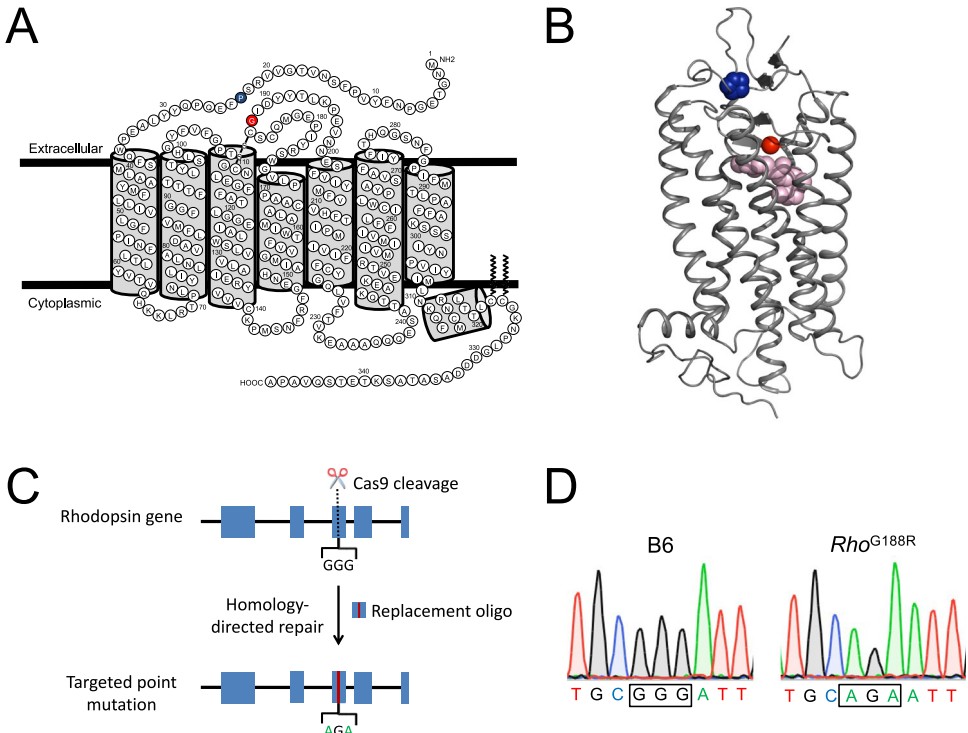

**Fig. 1 | P23H and G188R mutations in rhodopsin. A**, **B** Mutations highlighted on the structure of rhodopsin. Murine rhodopsin secondary structure (**A**) and crystal structure of bovine rhodopsin (**B**, PDB ID: 1U19) highlighting mutated proline (blue) and mutated glycine (red) residues examined in the current study. The chromophore 11-*cis* retinal is shown as pink spheres in the crystal structure. **C**, **D** Generation of G188R rhodopsin knockin mice by CRISPR/Cas9 gene targeting. Overview of

gene targeting strategy (**C**) and sequence in the region of the mutation (**D**) are shown. sgRNA was selected so that Cas9 would cut in exon 3 in the vicinity of glycine (GGG) at position 188. Homology-directed repair in the presence of a targeting oligo with the desired mutation introduces an arginine (AGA) mutation at position 188. Chromatograms in the region of the mutation are shown from sequencing of genomic DNA from B6 and *Rho*G188R mice (**D**).

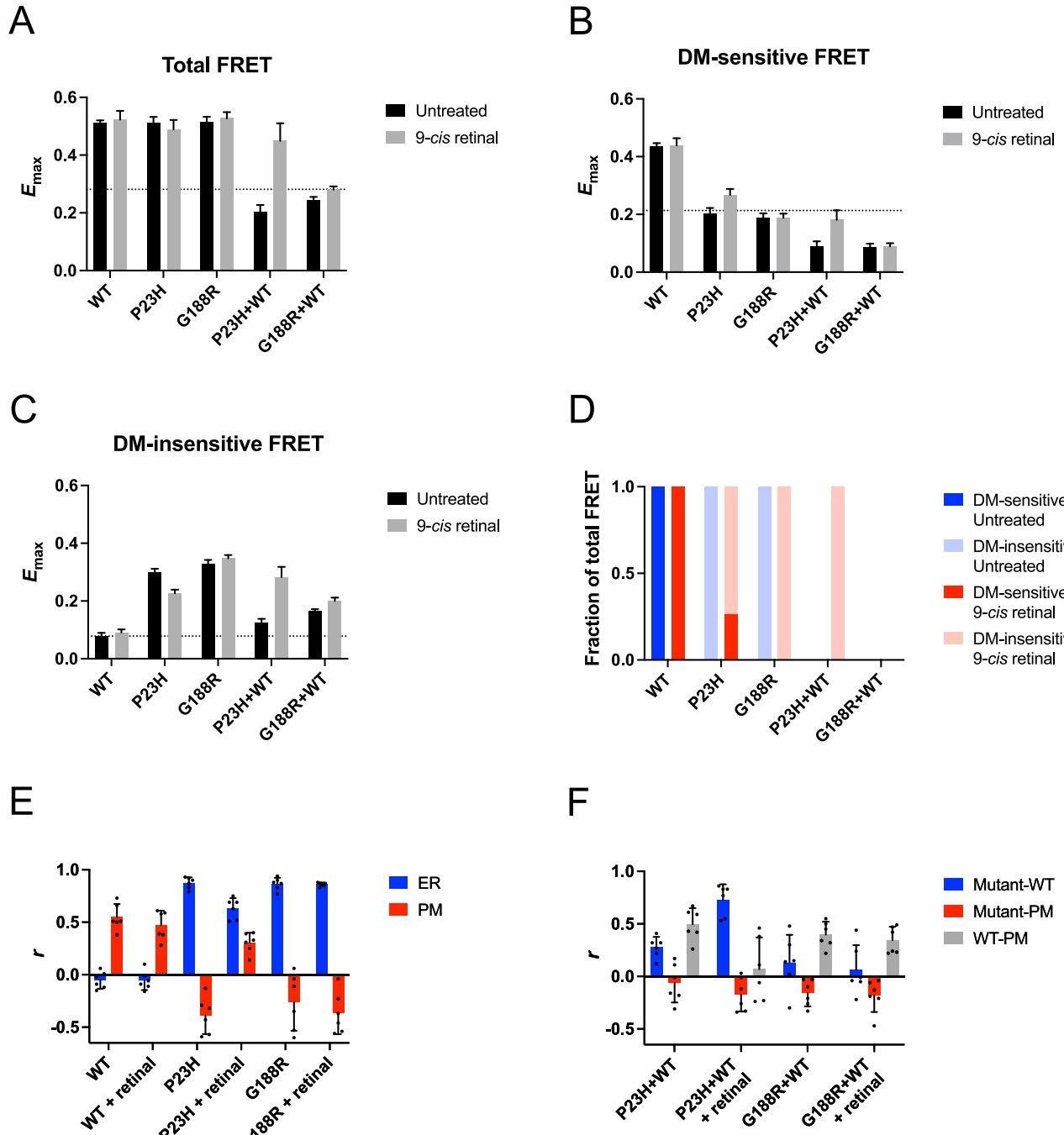

**Fig. 2 | Aggregation and mislocalization of mutant rhodopsin in HEK293 cells.**
**A–D** Summary of FRET analysis for WT, P23H, and G188R rhodopsin expressed alone and P23H or G188R rhodopsin coexpressed with WT rhodopsin. Cells were either untreated or treated with 15 μM 9-*cis* retinal. Total (**A**), DM-sensitive (**B**), and DM-insensitive (**C**) FRET $E_{max}$ values are plotted along with the standard error from fits of the FRET curves shown in Supplementary Fig. 1. The non-specific FRET $E_{max}$, defined previously[33], is indicated by the dashed lines. Fitted values and statistical analyses of the data are reported in Supplementary Tables 1 and 2. **D** The fraction of the total specific FRET signal derived from specific DM-sensitive (blue) and specific DM-insensitive FRET (red) is plotted. **E**, **F** Colocalization analysis. The Pearson's correlation coefficient (*r*) was computed from confocal microscopy images of rhodopsin in both untreated and 9-*cis* retinal-treated cells exhibited

singly expressed (**E**) or coexpressed rhodopsins (**F**) for each condition represented in Fig. 3. Individual data points and mean values are reported with the associated standard deviation (number of images, *n* = 6). **E** The Pearson's correlation coefficient is reported for comparisons of the fluorescence signal from YFP-tagged rhodopsin with the fluorescence signal from an ER marker (blue) or a plasma membrane (PM) marker (red). **F** The Pearson's correlation coefficient is reported for comparisons of fluorescence from mTq2-tagged WT and YFP-tagged mutant rhodopsin (blue), YFP-tagged mutant rhodopsin and a PM marker (red), and mTq2-tagged WT rhodopsin and a PM marker (gray). Source data are provided as a Source Data file.

rhodopsin in both untreated and 9-*cis* retinal-treated cells exhibited specific total FRET that consisted of only specific DM-sensitive FRET (Fig. 2A–D), indicative of the formation of oligomers. The formation of oligomers is consistent with the supramolecular organization of

rhodopsin in native rod outer segment (ROS) disc membranes[35]. The localization of rhodopsin within the cell was determined by colocalization analysis using plasma membrane and ER markers and computing the Pearson's correlation coefficient (*r*). WT rhodopsin under both

conditions colocalized with the plasma membrane marker (Figs. 2E and 3A), indicating that WT rhodopsin is properly targeted to the plasma membrane.

Both P23H and G188R mutant rhodopsin in untreated cells exhibited similar properties. In contrast to WT rhodopsin, both mutants exhibited specific total FRET derived from DM-insensitive FRET (Fig. 2A–D), indicative of the formation of aggregates rather than oligomers. Both mutants colocalized with the ER marker but not the plasma membrane marker, indicating that the mutants are retained in the ER (Figs. 2E and 3A). When cells were treated with 9-cis retinal, the two mutants displayed different properties. The specific total FRET for P23H rhodopsin derived from both DM-sensitive and DM-insensitive FRET, indicating that the mutant forms both aggregates and oligomers. The mutant colocalized with both the ER and plasma membrane markers (Figs. 2E and 3A). Thus, 9-cis retinal can rescue some P23H rhodopsin, allowing it to fold properly to form oligomers and be transported properly to the plasma membrane. In contrast, G188R rhodopsin in the presence of 9-cis retinal exhibited the same FRET and colocalization properties as the mutant in untreated cells, indicating that 9-cis retinal has no effect on this mutant.

To characterize potential effects in heterozygous mice coexpresssing mutant and WT rhodopsin, both mutants were coexpressed with WT rhodopsin in cells. In untreated cells, both the P23H and G188R mutants exhibited similar properties. Neither mutant exhibited specific total FRET (Fig. 2A), indicating the absence of physical interactions between the mutants and WT rhodopsin. Despite the absence of specific total FRET, both mutants did exhibit a small specific DM-insensitive FRET signal (Fig. 2C), which was shown previously to derive from aggregation between a small population of misfolded WT rhodopsin that may not occur in rod photoreceptor cells[36]. Consistent with these findings, some level of colocalization was present between both mutants and WT rhodopsin in the cell, but the mutants were mostly absent in the plasma membrane whereas WT rhodopsin colocalized with the plasma membrane marker as when expressed in the absence of the mutant (Figs. 2F and 3B). In cells treated with 9-cis retinal, both the FRET and colocalization properties in cells coexpressing G188R and WT rhodopsin were the same as those in untreated cells (Figs. 2 and 3B), which further supports the notion that 9-cis retinal has no effect on the G188R rhodopsin mutant. In contrast, treatment of cells coexpressing P23H and WT rhodopsin with 9-cis retinal resulted in different profiles compared to those in untreated cells. A specific total FRET signal was observed that was fully composed of specific DM-insensitive FRET (Fig. 2A–D), indicating that P23H rhodopsin forms aggregates with WT rhodopsin. Accordingly, colocalization analysis indicated colocalization of P23H and WT rhodopsin along with a diminishment of the colocalization of WT rhodopsin in the plasma membrane (Figs. 2F and 3B).

A summary of the aggregation profiles of murine P23H and G188R mutant rhodopsins is presented in Supplementary Table 3. These aggregation properties of the murine form of the mutants largely mirror those of their human counterparts characterized previously[33], which demonstrates that the behavior of these mutants in mouse models is predicted to be similar to those in human patients. The consequences of these mutations characterized in vitro were next examined in vivo in mouse models.

### Retinal degeneration in mouse models of adRP

Mouse models of adRP examined here include a P23H rhodopsin knockin mouse, previously generated and characterized[20], and a G188R rhodopsin knockin mouse generated here by CRISPR/Cas9 technology (Fig. 1C, D). The retina in mice that were either heterozygous ($Rho^{P23H/+}$ or $Rho^{G188R/+}$) or homozygous ($Rho^{P23H}$ or $Rho^{G188R}$) for the mutant rhodopsins were characterized in parallel with that in C57Bl/6J (B6) mice. Retinal degeneration was characterized in different aged mice by quantifying the number of nuclei spanning the outer

nuclear layer, which corresponds to the nuclei of photoreceptor cells. Spider plots were generated to determine the retinal degeneration across the retina at different distances from the optic nerve (Fig. 4A–D). The kinetics of photoreceptor cell loss was exponential and defined by computing the rate constant ($k$) (Fig. 4E, F). At 2 weeks of age, $Rho^{P23H/+}$ and $Rho^{G188R/+}$ mice had similar numbers of nuclei, exhibiting minimal photoreceptor cell loss. The difference in retinal degeneration between $Rho^{P23H/+}$ and $Rho^{G188R/+}$ mice became progressively larger with age. The rate of photoreceptor cell loss in $Rho^{G188R/+}$ mice was 2 times faster than that in $Rho^{P23H/+}$ mice (Table 1). In both heterozygous mutant mice, photoreceptor cell loss was greater on the inferior region compared to the superior region of the retina (Fig. 4B–D). The rate of photoreceptor cell loss in the inferior region of the retina in heterozygous mutant mice was 60–70% faster compared to that occurring on the superior region of the retina (Table 1). Thus, both $Rho^{P23H/+}$ and $Rho^{G188R/+}$ mice exhibit a more severe retinal degeneration in the inferior retina.

Homozygous $Rho^{P23H}$ and $Rho^{G188R}$ mice exhibited a more severe retinal degeneration compared to their heterozygous counterparts. The rate of photoreceptor cell loss in $Rho^{P23H}$ and $Rho^{G188R}$ mice compared to their heterozygous counterparts was 7–8 times faster in the superior region of the retina and 4 times faster in the inferior region of the retina (Table 1). Even at 2 weeks of age, both homozygous mutant mice exhibited significant photoreceptor cell loss, with the loss in $Rho^{G188R}$ mice being greater than that in $Rho^{P23H}$ mice (Fig. 4A). Similar to heterozygous mutant mice, the rate of photoreceptor cell loss in $Rho^{G188R}$ mice was 2 times faster than that in $Rho^{P23H}$ mice (Table 1). In contrast to heterozygous mutant mice, the inferior retina in homozygous mutant mice did not exhibit a more severe retinal degeneration compared to the superior retina. In fact, $Rho^{P23H}$ mice exhibited a more severe retinal degeneration in the superior retina with a 19 % faster rate of photoreceptor cell loss compared to that in the inferior retina (Table 1). $Rho^{G188R}$ mice exhibited similar rates of photoreceptor cell loss in the inferior and superior regions of the retina.

The functional impact of the observed retinal degeneration was characterized by electroretinography (ERG) in 1-month-old mice. The scotopic a-wave directly reflects the function of rod photoreceptor cells[37,38]. The b-wave largely reflects the function of bipolar cells and therefore the scotopic b-wave indirectly reflects all photoreceptor cell function and photopic b-wave indirectly reflects cone photoreceptor cell function[39]. The maximal amplitude ($R_{max}$) of the scotopic a-wave was diminished in both $Rho^{P23H/+}$ and $Rho^{G188R/+}$ mice, with a greater deficit in the latter (Fig. 4G and Supplementary Tables 4 and 5). No difference was observed in the $K_A$ (intensity generating half-maximal amplitude) of these curves. A reduction in $R_{max}$ with no change in $K_A$ indicates the loss of rod photoreceptor cells with phototransduction capabilities being maintained in the remaining rod photoreceptor cells[37]. The $R_{max}$ of the scotopic b-wave was similar for B6 and $Rho^{P23H/+}$ mice but that of $Rho^{G188R/+}$ mice was significantly reduced (Fig. 4H and Supplementary Tables 4 and 5). Despite the loss of rod photoreceptor cells in $Rho^{P23H/+}$ mice, a significant change in the $R_{max}$ of the scotopic b-wave was not observed, which is due to the increased sensitivity of rod bipolar cells that accompanies the mild retinal degeneration at this age in this model[40]. The significant reduction in the $R_{max}$ of the scotopic b-wave in $Rho^{G188R/+}$ mice is indicative of greater rod photoreceptor cell loss in these mice compared to $Rho^{P23H/+}$ mice. The $R_{max}$ of the photopic b-wave was unaffected in $Rho^{P23H/+}$ mice but was diminished in $Rho^{G188R/+}$ mice (Fig. 4I and Supplementary Tables 4 and 5), which indicates that cone photoreceptor cells appear to be unaffected in $Rho^{P23H/+}$ mice but may be beginning to degenerate in $Rho^{G188R/+}$ mice. Nonetheless, the change in $R_{max}$ of the photopic b-wave was less than that of the scotopic b-wave for $Rho^{G188R/+}$ mice, indicating that at this time point, the effect on cone function was less than the effect on rod function. Both histological and functional characterizations indicate

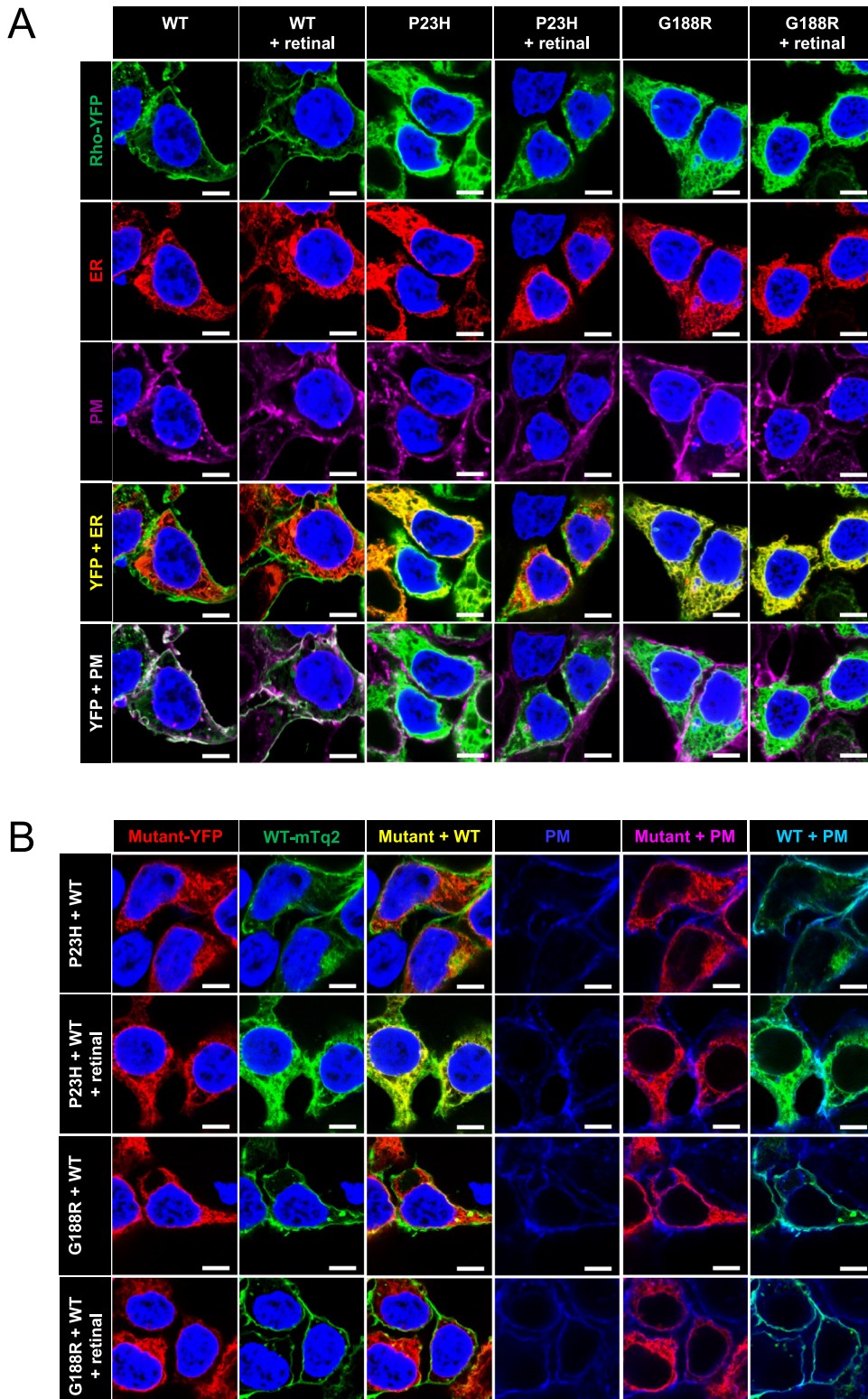

**Fig. 3 | Mislocalization of mutant rhodopsin expressed in HEK293 cells that were untreated or treated with 15 μM 9-*cis* retinal. A** Singly expressed rhodopsin. Each row shows images of fluorescence from YFP-tagged WT, P23H, or G188R rhodopsin (green), the ER marker DsRed2-ER (red), the plasma membrane (PM) marker WGA (magenta), overlays of the fluorescence from YFP-tagged rhodopsin and the ER marker (yellow), and overlays of the fluorescence from YFP-tagged rhodopsin and the PM marker (white). **B** HEK293 cells coexpressing mTq2-tagged WT rhodopsin and either YFP-tagged P23H or G188R rhodopsin. Each column shows images of fluorescence from YFP-tagged mutant rhodopsin (red), mTq2-tagged WT rhodopsin (green), overlays of the fluorescence from YFP-tagged mutant rhodopsin and mTq2-tagged WT rhodopsin (yellow), PM marker (blue), overlays of fluorescence from YFP-tagged mutant rhodopsin and the PM marker (magenta), and overlays of fluorescence from mTq2-tagged WT rhodopsin and the PM marker (cyan). Nuclei were stained by DAPI (blue) and were shown in all images except the last 3 columns in **B**. Scale bar, 5 μm. Images are representative of at least three different experiments.

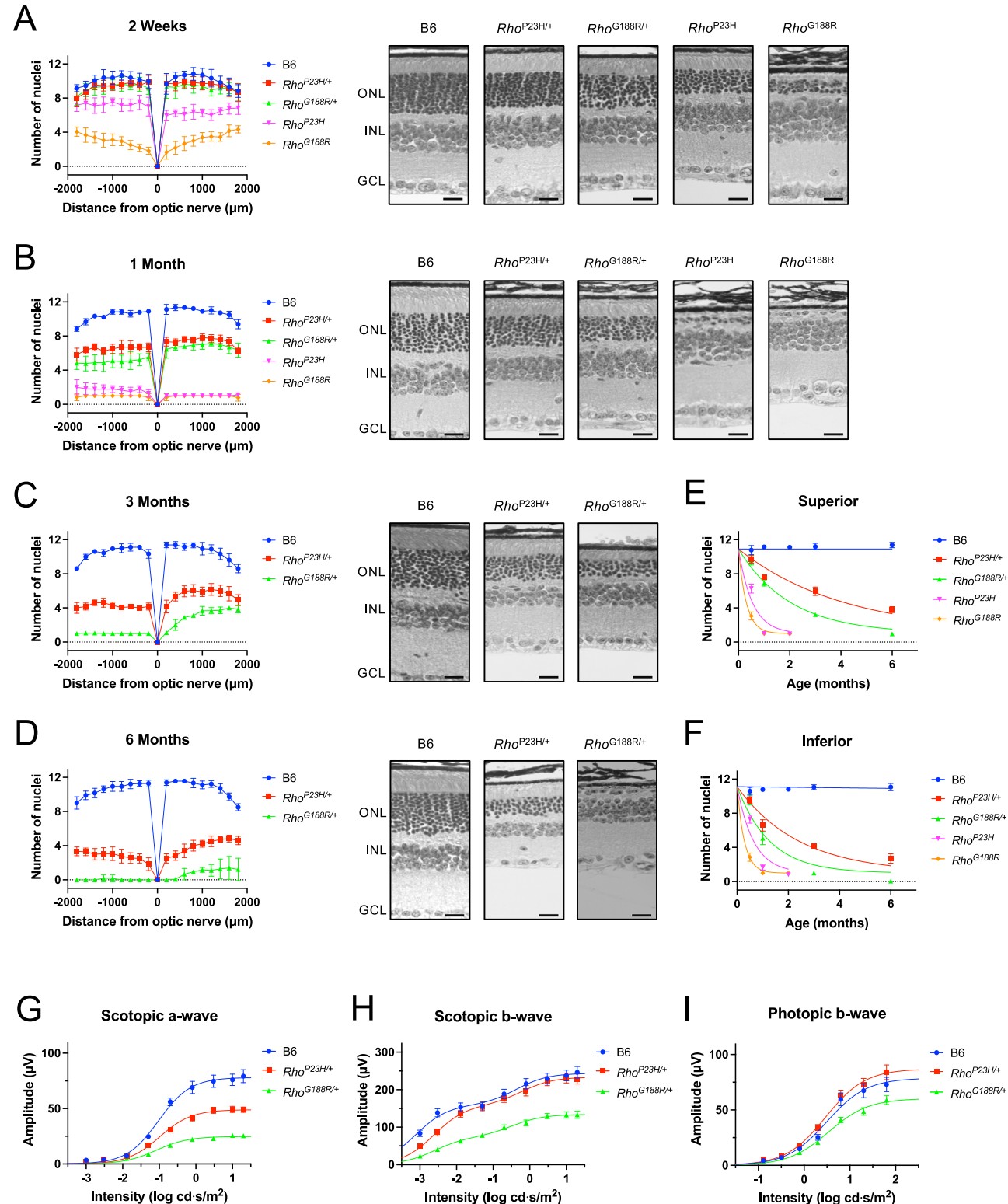

that the G188R mutation leads to a more severe retinal degeneration phenotype compared to that promoted by the P23H mutation.

## Reduced expression and mislocalization of rhodopsin in the retina of mutant mice

In in vitro studies, the cDNA encoding the rhodopsin mutants was used to transfect cells and therefore transcription-related effects due to the mutations are not examined. The pathogenic effect of point mutations in the rhodopsin gene can be at the level of the transcript and/or

protein[41]. Previously, the effect of the P23H mutation was shown to be independent of effects related to transcription since similar levels of rhodopsin transcripts were present regardless of the presence of the mutation, at least in heterozygous mutant mice[20,42]. To determine if the point mutations affect the level of transcripts, RT-qPCR was conducted on retinal samples from mice. Samples were collected from mice at 2 weeks of age and the level of rhodopsin transcripts was normalized to that of 18s rRNA or transducin (*Gnat1*) to account for any retinal degeneration (Fig. 5A). In both *Rho*^P23H/+ and *Rho*^G188R/+ mice, the level of

**Fig. 4 | Progression of retinal degeneration promoted by P23H and G188R rhodopsin. A**–**D** Retinal sections were prepared from mice that were 2 weeks (**A**), 1 month (**B**), 3 months (**C**), or 6 months (**D**) of age. Spider plots (left side) showing the number of nuclei spanning the outer nuclear layer at different distances from the optic nerve in the superior (positive) or inferior (negative) retina are shown. Mean values along with the standard deviation are reported (number of mice, $n = 6$). The right-side shows images of the retina from the inferior region with the outer nuclear layer (ONL), inner nuclear layer (INL), and ganglion cell layer (GCL) indicated. Scale bar, 25 μm. **E**, **F** The number of nuclei spanning the outer nuclear layer at 600–1000 μm from the optic nerve in the superior (**E**) or inferior (**F**) region of the retina in mice of different ages was plotted to determine the kinetics of

retinal degeneration. The data were fit with an exponential equation for one-phase decay by non-linear regression to determine the rate constant ($k$), which is reported in Table 1. Mean values along with the standard deviation were plotted (number of mice, $n = 6$). **G**–**I** The amplitude of the a-wave (**G**) and b-wave (**H**) in scotopic ERG responses and the amplitude of the b-wave in photopic ERG responses (**I**) were recorded at increasing intensities of light. Mean values are plotted with the standard error (number of mice, B6, $n = 15$; $Rho^{P23H/+}$, $n = 13$; $Rho^{G188R/+}$, $n = 14$). Data were fit to dose-response models as described in the Methods. Fitted values and statistical analyses are reported in Supplementary Tables 4 and 5. Source data are provided as a Source Data file.

rhodopsin transcripts was comparable to that in B6 mice regardless of whether transcripts were normalized to that of 18s rRNA or transducin. Thus, the mutations do not impact transcription and similarities in transcript levels when normalized to 18s rRNA and transducin are consistent with the minimal retinal degeneration occurring at this age (Fig. 4A). Both homozygous mutant mice exhibited reduced levels of rhodopsin transcripts when normalized to 18S rRNA compared to that in B6 mice, which reflects the retinal degeneration occurring even in young mice. Rhodopsin transcript levels were reduced even when normalized to that of transducin, indicating that once a significant level of retinal degeneration occurs, transcription may become affected.

Although rhodopsin transcript levels are comparable in $Rho^{P23H/+}$ and $Rho^{G188R/+}$ mice to that in B6 mice at 2 weeks of age, the levels of rhodopsin protein in these mice were significantly reduced. Rhodopsin protein levels were estimated by quantifying bands corresponding to rhodopsin in Western blots of retinal extracts from 2-week-old mutant mice (Fig. 5B, C). The amount of rhodopsin present in the retina of $Rho^{P23H/+}$ and $Rho^{G188R/+}$ mice was about half of that in B6 mice (Fig. 5D). The pattern of bands in Western blots for both heterozygous mutant mice was similar to that for B6 mice, with the band corresponding to monomeric rhodopsin being the predominant band (Fig. 5B). In contrast to heterozygous mutant mice, Western blots of $Rho^{P23H}$ and $Rho^{G188R}$ mice exhibited a pattern of bands where the monomeric band was absent and only bands corresponding to multiples of rhodopsin were present (Fig. 5C). In addition, the level of rhodopsin detected in Western blots of retinal samples from $Rho^{P23H}$ and $Rho^{G188R}$ mice was only 2% of that from B6 mice (Fig. 5D). Thus, it appears that most of the rhodopsin mutants are degraded and that bands resolved in Western blots from retinal extracts of $Rho^{P23H/+}$ and $Rho^{G188R/+}$ mice represent mostly WT rhodopsin.

The localization of rhodopsin in photoreceptor cells was examined in retinas from 2-week-old mice (Fig. 5E), an early time point where rhodopsin and ROS discs are beginning to form at a rapid rate in B6 mice[43,44] and retinal degeneration is minimal in heterozygous mutant mice, and in 1-month-old mice (Fig. 5F), where rhodopsin expression and ROS length have stabilized in B6 mice[43,44] and retinal degeneration is significant. Rhodopsin was stained with an anti-4D2 antibody, which detects the amino-terminal region of rhodopsin[45], and an anti-1D4 antibody, which detects an epitope in the carboxy-terminal tail of rhodopsin[46]. The staining of anti-4D2 and anti-1D4 antibodies was exclusively in the ROS in B6 mice at both ages (Fig. 5E, F), demonstrating the proper localization of rhodopsin within the photoreceptor cell. The two antibodies stained differently in $Rho^{P23H/+}$ and

$Rho^{G188R/+}$ mice at both ages of mice. The anti-4D2 antibody stained both the ROS and the outer nuclear layer, indicating mislocalization of some of the receptors. The outer segments were shorter in the heterozygous mutant mice compared to B6 mice. In contrast, the anti-1D4 antibody only stained the ROS of heterozygous mutant mice. When an antigen retrieval step was performed, the anti-1D4 antibody stained similarly as the anti-4D2 antibody. Protein aggregation can prevent access of antibodies to an epitope in fixed tissue[47]. Thus, one possible explanation for the masking of the 1D4 epitope in the mutant mice may be related to rhodopsin aggregation.

In 2-week-old $Rho^{P23H}$ and $Rho^{G188R}$ mice (Fig. 5E), anti-4D2 antibody staining was exclusively in the outer nuclear layer. In contrast to heterozygous mutant mice, anti-1D4 antibody staining was observed in the absence of antigen retrieval, however, the level of staining was sporadic and not as extensive as anti-4D2 antibody staining. Antigen retrieval was required for similar levels of staining to that of anti-4D2 antibody staining. The nature of rhodopsin stained by the anti-1D4 antibody in homozygous mutant mice is unclear, but it is distinct from that detected after antigen retrieval. Most of the rod photoreceptor cells are lost by 1 month of age in homozygous mutant mice. Thus, neither anti-4D2 nor anti-1D4 antibody staining is observed except for some sporadic staining by the anti-4D2 antibody in $Rho^{P23H}$ mice (Fig. 5F). Taken together, both P23H and G188R mutant rhodopsins that are not degraded appear to mislocalize in rod photoreceptor cells as is observed when they are heterologously expressed in HEK293 cells.

## Altered rhodopsin packing in ROS disc membranes of mutant mice

In vitro characterizations indicate that mutant rhodopsin does not aggregate with WT rhodopsin or disrupt its normal localization (Fig. 2). Thus, in mice when both WT and mutant rhodopsin are expressed, WT rhodopsin should traffic normally to the ROS and incorporate into ROS disc membranes. Proper localization of at least some rhodopsin to the ROS in $Rho^{P23H/+}$ and $Rho^{G188R/+}$ mice is observed (Fig. 5E, F) and is likely the WT form since most of the mutants appear to be degraded (Fig. 5D). The structure of the ROS and disc membranes were examined from $Rho^{P23H/+}$ and $Rho^{G188R/+}$ mice by atomic force microscopy (AFM) and electron microscopy (EM) to determine if rhodopsin is properly packed into the disc membrane of the ROS. ROS disc membranes prepared from $Rho^{P23H/+}$ and $Rho^{G188R/+}$ mice were examined by AFM. Samples were prepared from young mice that were 4 weeks old, an age where the rate of synthesis of the ROS has stabilized and full adult length achieved[43] and where the impact of retinal degeneration is reduced allowing preparation of sufficient quantities of ROS disc membranes. The quality of ROS disc membranes from both mutant mice were lower than that typically observed in B6 mice, likely due to retinal degeneration. In B6 mice, ROS disc membranes display a well-structured rim region and a lamellar region densely packed with nanodomains of oligomeric rhodopsin (Fig. 6A)[48–52]. ROS disc membranes exhibiting nanodomains of rhodopsin in the lamellar region were also observed in samples from both $Rho^{P23H/+}$ and $Rho^{G188R/+}$ mice (Fig. 6A). In $Rho^{G188R/+}$ mice, however, properly formed ROS disc

### Table 1 | Kinetics of photoreceptor cell loss

| Region of retina | $k$ (month⁻¹) | | | | |
|---|---|---|---|---|---|
| | B6 | $Rho^{P23H/+}$ | $Rho^{G188R/+}$ | $Rho^{P23H}$ | $Rho^{G188R}$ |
| Superior | $4.1 \times 10^{-9} \pm 0.006$ | $0.24 \pm 0.02$ | $0.49 \pm 0.03$ | $1.84 \pm 0.12$ | $3.29 \pm 0.29$ |
| Inferior | $3.4 \times 10^{-3} \pm 0.008$ | $0.41 \pm 0.03$ | $0.78 \pm 0.07$ | $1.55 \pm 0.12$ | $3.51 \pm 0.42$ |

Fitted parameters from Fig. 4E, F are shown with the standard errors.

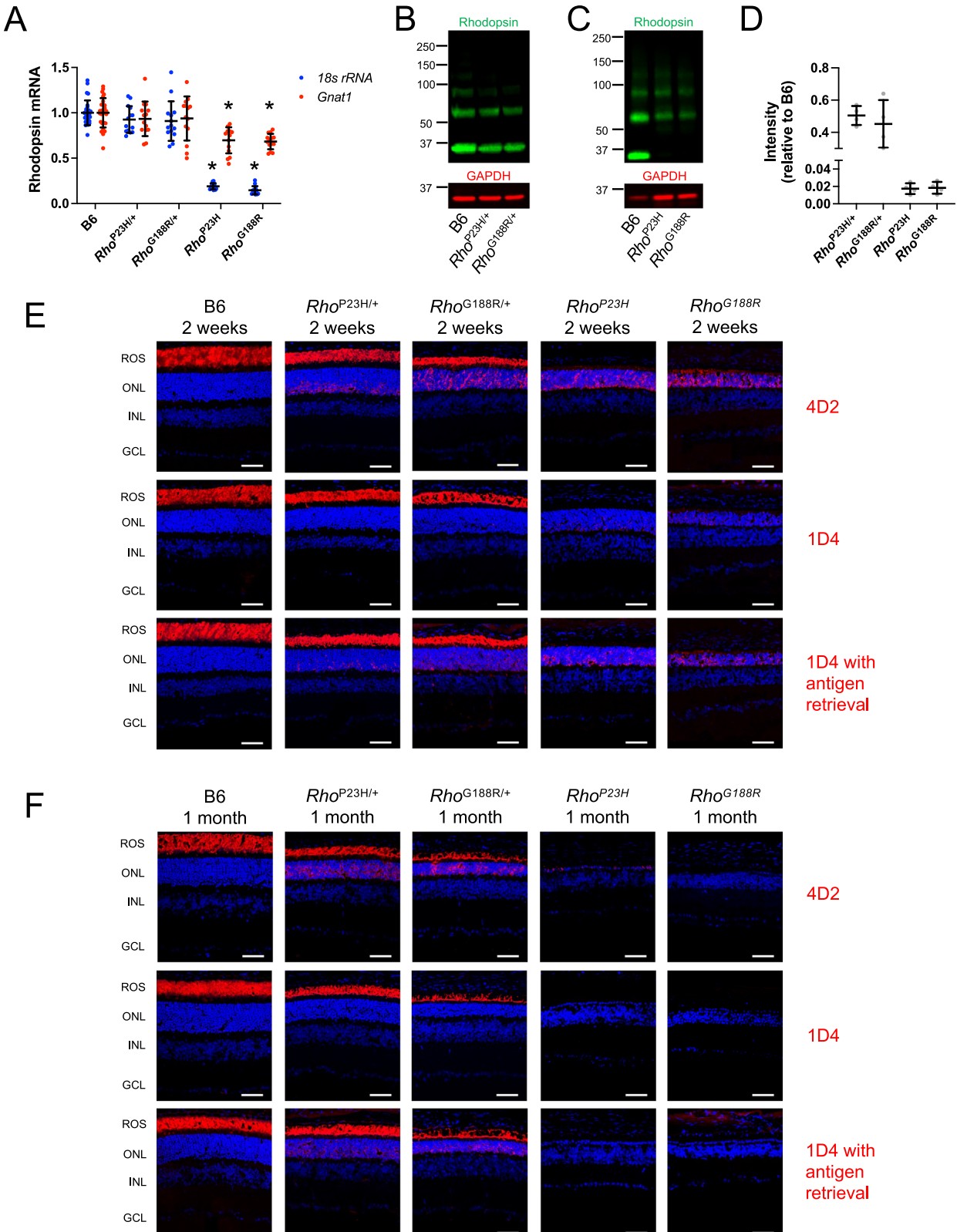

membranes were only infrequently observed, which is consistent with the more severe retinal degeneration occurring in these mice compared to that in $Rho^{P23H/+}$ mice (Fig. 4).

The generally lower quality of ROS disc membranes observed by AFM in $Rho^{P23H/+}$ and $Rho^{G188R/+}$ mice was corroborated by EM of retinal sections (Fig. 6B). The ROS of B6 mice were long and displayed well-ordered stacking of disc membranes. In contrast, the ROS of $Rho^{P23H/+}$ and $Rho^{G188R/+}$ mice were shorter, which was also observed by immunohistochemistry (Fig. 5F), and fewer in number. $Rho^{P23H/+}$ mice exhibited more ROS compared to $Rho^{G188R/+}$ mice and there were more ROS exhibiting stacked discs, although they were not as well structured as those in the ROS of B6 mice. The effect of retinal degeneration was evident in both $Rho^{P23H/+}$ and $Rho^{G188R/+}$ mice, with it being more evident in the latter.

**Fig. 5 | Expression and mislocalization of rhodopsin mutants in the retina of mutant mice. A** RT-qPCR was conducted on retinal samples from 2-week-old mice to determine the level of rhodopsin transcripts normalized to *18s rRNA* (blue) or *Gnat1* (red) transcripts. Individual data points are reported relative to that of control samples from B6 mice (number of mice, *n* = 28 for B6 and *n* = 14 for all others). The mean and standard deviation are indicated. Statistical analyses are reported in Supplementary Table 6 and statistically significant differences are indicated (*). **B**, **C** Western blots of retinal extracts from 2-week-old B6, *Rho*[P23H/+], and *Rho*[G188R/+] mice (**B**) or from 2-week-old B6, *Rho*[P23H], and *Rho*[G188R] mice (**C**). Rhodopsin was detected by the anti-1D4 antibody (green) and GAPDH was detected with an anti-GAPDH antibody (red), which served as a loading control. Molecular weights of the marker are shown in kDa. **D** Rhodopsin expression levels were estimated by quantifying intensities of bands on Western blots. Rhodopsin band intensities were summed and normalized to that of GAPDH. The intensity of bands corresponding to rhodopsin in retinal extracts from mutant mice are shown relative to that of B6 mice. Individual data points (gray) are presented along with the mean and standard deviation (number of mice, *n* = 4). **E**, **F** Rhodopsin was labeled in retinal cryosections from 2-week- (**E**) or 1-month-old (**F**) mice with the anti-4D2 antibody, anti-1D4 antibody, or anti-1D4 antibody after antigen retrieval (red). Nuclei were labeled with DAPI (blue). The ROS, outer nuclear layer (ONL), inner nuclear layer (INL), and ganglion cell layer (GCL) are labeled. Scale bar, 50 µm. Images are representative of at least three different experiments. Source data are provided as a Source Data file.

AFM image analysis of ROS disc membranes was performed only on samples from *Rho*[P23H/+] mice since enough ROS disc membranes of sufficient quality could not be obtained from *Rho*[G188R/+] mice. Analysis of AFM images of ROS disc membranes provides detailed information about the size of discs and the packing of rhodopsin within the ROS disc membrane[53,54]. It is unknown the amount of mutant P23H rhodopsin that accompanies WT rhodopsin to the ROS. If none of the mutant is present in the ROS, then only half the amount of rhodopsin is expected to be present in the ROS of *Rho*[P23H/+] mice as that in B6 mice. Only half the amount of rhodopsin is expressed in heterozygous rhodopsin knockout (*Rho*[+/-]) mice compared to B6 mice[55]. The properties of ROS disc membranes from *Rho*[P23H/+] mice were compared to those of both B6 and *Rho*[+/-] mice, which were previously characterized[48]. The ROS disc membrane properties of *Rho*[P23H/+] mice displayed both similarities and differences from those of both B6 and *Rho*[+/-] mice (Fig. 6C–H). The size of ROS discs was similar among the different mice (Fig. 6C). The size of nanodomains formed by oligomeric rhodopsin was smaller in *Rho*[P23H/+] mice than those formed in both B6 and *Rho*[+/-] mice (Fig. 6D), perhaps indicating a change to the equilibria defining the oligomeric status of the receptor[35,56]. The number and density of nanodomains in *Rho*[P23H/+] mice was similar to that in B6 mice but greater than that in *Rho*[+/-] mice (Fig. 6E, F). The number of rhodopsin packed into ROS disc membranes in *Rho*[P23H/+] mice was similar to that in *Rho*[+/-] mice but lower than that in B6 mice (Fig. 6G). The density of rhodopsin in ROS disc membranes in *Rho*[P23H/+] mice was in between that in B6 and *Rho*[+/-] mice (Fig. 6H). The packing of rhodopsin in ROS disc membranes of *Rho*[P23H/+] mice was altered compared to that in both B6 and *Rho*[+/-] mice, indicating that the packaging of rhodopsin into the disc membranes is perturbed.

## Relationship between PROTEOSTAT and TUNEL staining in photoreceptor cells

In in vitro studies, mislocalization of rhodopsin is accompanied by aggregation of the receptor. To examine the possibility that rhodopsin aggregation can contribute to photoreceptor cell death, retinal cryosections were stained with PROTEOSTAT, which is a molecular rotor dye that becomes fluorescent upon binding to aggregated proteins and was used previously to detect mutant rhodopsin aggregates in the retina[17,57]. Photoreceptor cell death was assessed by terminal deoxynucleotidyl transferase dUTP nick end labeling (TUNEL). B6 mice did not display TUNEL or PROTEOSTAT positive cells in the outer nuclear layer (Fig. 7A), indicating the absence of photoreceptor cell death and aggregation. Some PROTEOSTAT staining was detected throughout the retina of B6 mice and may represent non-specific staining as the level of fluorescence was variable in different experiments. This staining in B6 mice was not considered to be an indicator of aggregation. In contrast to B6 mice, both heterozygous and homozygous mutant mice displayed TUNEL and distinct PROTEOSTAT staining in the outer nuclear layer (Fig. 7A).

The relationship between TUNEL and PROTEOSTAT staining in the retina was examined at 2 weeks, 3 weeks, and 1 month of age for heterozygous mutant mice and at 2 weeks of age for homozygous mutant mice. No TUNEL or PROTEOSTAT staining was observed at these ages in B6 mice. The level of TUNEL-positive cells in mutant mice reflected the severity of retinal degeneration observed in these mice. *Rho*[P23H/+] mice had the greatest number of TUNEL-positive cells at 3 weeks of age whereas *Rho*[G188R/+] mice exhibited the greatest number of TUNEL-positive cells at 2 weeks of age and decreased numbers as mice aged (Fig. 7B, C). The trends in time course of cell death were the same in both superior and inferior regions of the retina. The peak of cell death occurring at 3 weeks of age in *Rho*[P23H/+] mice is consistent with a previous study[58]. The earlier peak of cell death occurring in *Rho*[G188R/+] mice is consistent with the faster rate of photoreceptor cell loss observed compared to that in *Rho*[P23H/+] mice (Table 1). Homozygous mutant mice exhibited a greater number of TUNEL positive cells in the outer nuclear layer compared to heterozygous mutant mice at 2 weeks of age, with *Rho*[G188R] mice exhibiting greater cell death compared to *Rho*[P23H] mice. This trend is consistent with the more severe retinal degeneration observed in homozygous mutant mice that occurs earlier on compared to heterozygous mutant mice.

PROTEOSTAT staining was performed in samples from the same aged mice examined in the TUNEL assay to assess a possible relationship between aggregation and photoreceptor cell death. Indeed, the pattern of PROTEOSTAT-positive cells mirrored that of TUNEL-positive cells where the level of PROTEOSTAT-positive cells peaked at 3 weeks of age in *Rho*[P23H/+] mice and earlier in *Rho*[G188R/+] mice (Fig. 7B, C). There were also more PROTEOSTAT-positive cells in homozygous mutant mice compared to heterozygous mutant mice. This correlation between the levels of PROTEOSTAT and TUNEL positive cells points to a potential relationship between aggregation and photoreceptor cell death.

PROTEOSTAT staining in the outer nuclear layer was characterized further to better understand its nature and origin. To examine the possibility that PROTEOSTAT staining detects species unrelated to mutant rhodopsin aggregation, two controls were conducted. Rhodopsin knockout (*Rho*[-/-]) mice do not express rhodopsin and retinal degeneration occurs because of the critical role rhodopsin plays in maintaining photoreceptor cell structure[59]. No rhodopsin was detected in the retina of these mice and photoreceptor cell death was detected by TUNEL (Fig. 8A). No PROTEOSTAT staining was detected in the outer nuclear layer of the retina in *Rho*[-/-] mice. Thus, retinal degeneration does not promote the aggregation of non-rhodopsin proteins or any other species that can be stained by PROTEOSTAT. *Prph2*[Rd2] mice express peripherin 2, which is a structural protein required for the formation of ROS discs, with the *Rd2* mutation that causes a slow retinal degeneration phenotype[60]. Rhodopsin was mislocalized to the outer nuclear layer because of disruptions to ROS disc biogenesis and photoreceptor cell death was detected by TUNEL (Fig. 8A). No PROTEOSTAT staining was detected in the outer nuclear layer of the retina in *Prph2*[Rd2] mice. Thus, neither mislocalization of WT rhodopsin nor retinal degeneration causes the formation of aggregates detectable by PROTEOSTAT in the outer nuclear layer.

PROTEOSTAT used in the current study is advertised as a detection reagent for aggresomes, which are microtubule-dependent inclusion bodies containing large, aggregated proteins that can be observed by EM[61,62]. Misfolding rhodopsin

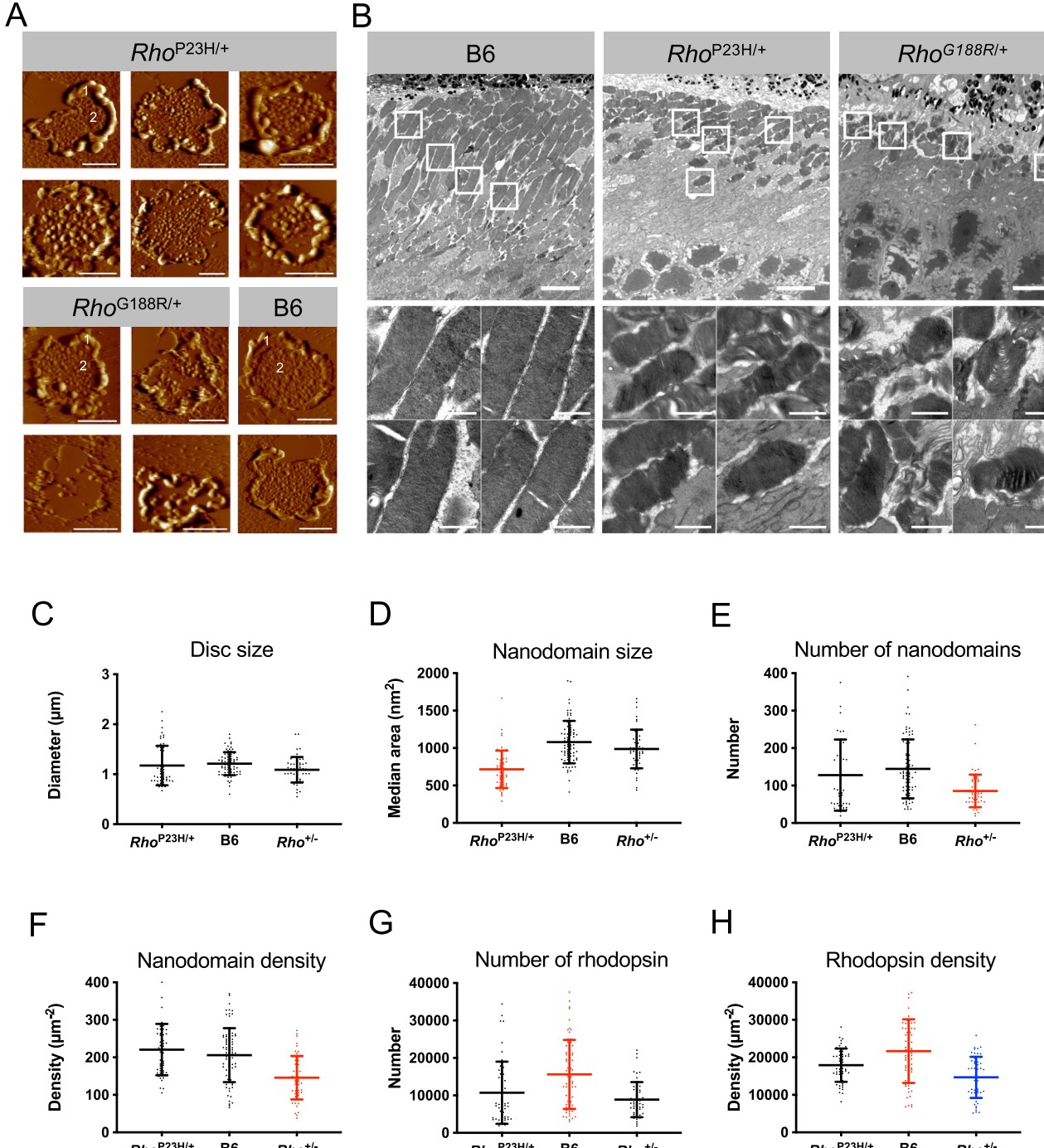

**Fig. 6 | ROS discs form in *Rho*^P23H/+ and *Rho*^G188R/+ mice and contain rhodopsin nanodomains. A** Sample of AFM images of ROS disc membranes from 4-week-old *Rho*^P23H/+ and *Rho*^G188R/+ mice. AFM images of ROS disc membranes from B6 mice are also provided as a reference. Scale bar, 500 nm. The rim region (1) and lamellar region containing rhodopsin nanodomains (2) are highlighted. **B** EM images of thin sections of retina from 1-month-old B6, *Rho*^P23H/+, and *Rho*^G188R/+ mice. The top images are lower magnification (scale bar, 5 µm) to show the overall retina and the lower images are zoomed in images (scale bar, 1 µm) of ROS in the regions marked by white boxes in the lower magnification images. Images are representative of at least 3 different experiments. **C–H** ROS disc membrane properties from AFM

analysis of samples from 4-week-old *Rho*^P23H/+, B6, and *Rho*^+/− mice. Mean values are reported with the standard deviation for disc diameter (**C**), median nanodomain size (**D**), number of nanodomains (**E**), nanodomain density (**F**), number of rhodopsin (**G**), and rhodopsin density (**H**). Data presented in the figure are also reported in Supplementary Table 7. Data for B6 and *Rho*^+/− mice are included as a reference and were those that were reported previously[48]. Mice exhibiting statistically significant differences ($P < 0.05$) are indicated by different coloring of the data. The same coloring of the data indicates that the differences among those data are not statistically significant ($P > 0.05$). Results of statistical analyses are reported in Supplementary Table 8. Source data are provided as a Source Data file.

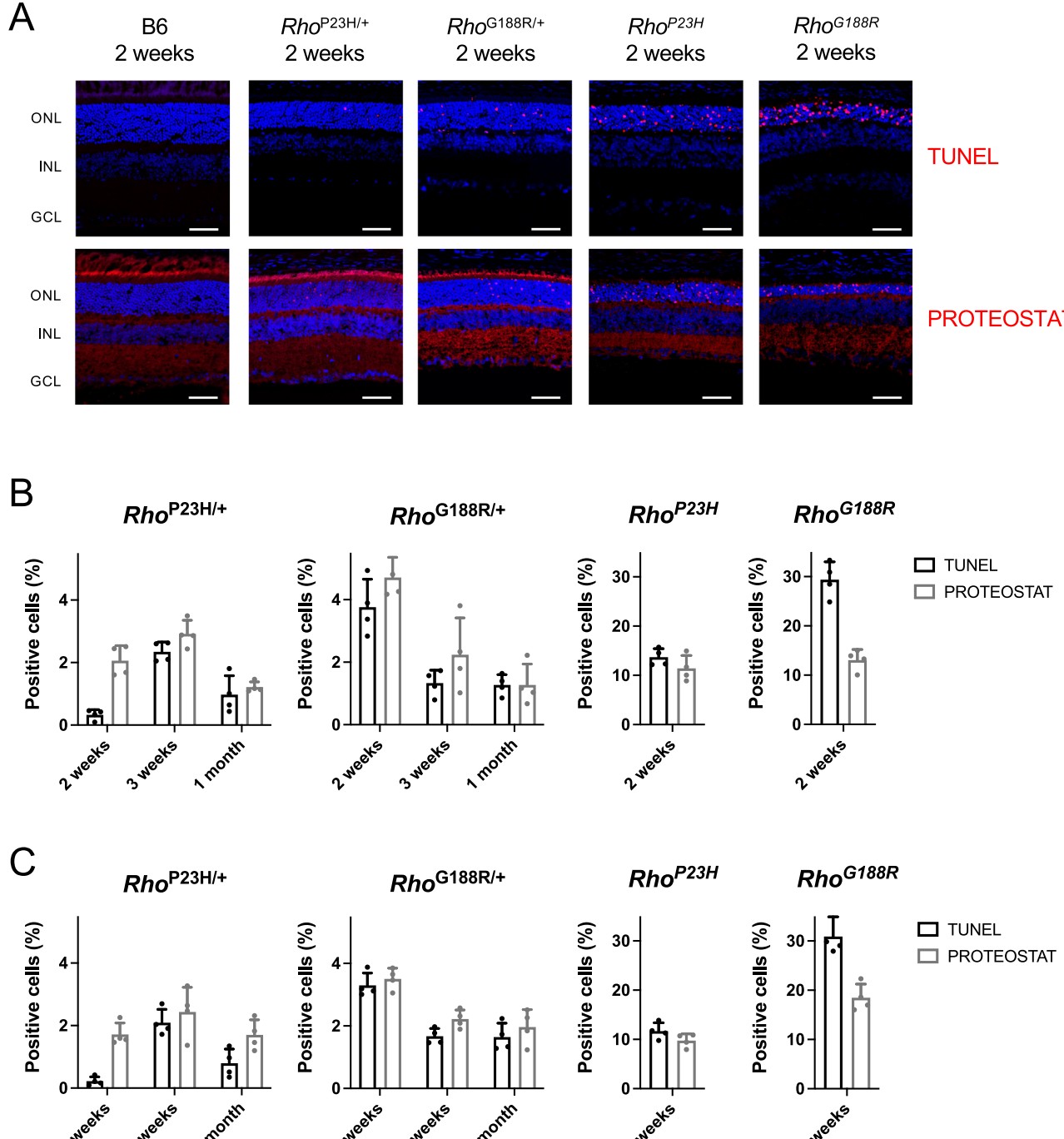

**Fig. 7 | Correlation between TUNEL and PROTEOSTAT staining in the outer nuclear layer. A** Retinal cryosections from 2-week-old mice were labeled by TUNEL or PROTEOSTAT (red). Nuclei were labeled with DAPI (blue). The ROS, outer nuclear layer (ONL), inner nuclear layer (INL), and ganglion cell layer (GCL) are labeled. Scale bar, 50 µm. **B, C** TUNEL and PROTEOSTAT positive cells in the outer nuclear layer were quantified in the superior (**B**) and inferior (**C**) regions of the retina of mutant mice. Individual data points along with the mean and standard deviation are shown (number of mice, $n = 4$). Source data are provided as a Source Data file.

mutants have previously been shown to form aggresomes in vitro[30,31]. Retinal cryosections were labeled with an anti-ubiquitin antibody to determine if PROTEOSTAT colocalizes with ubiquitin, a marker for aggresomes[61,63]. Only a subset of PROTEOSTAT staining in the outer nuclear layer colocalized with ubiquitin and structures exhibiting colocalization were not associated with nuclei (Fig. 8B), indicating that PROTEOSTAT is not detecting aggresomes. Confocal microscopy was conducted at higher magnification to examine the morphology of PROTEOSTAT

staining in the outer nuclear layer. Mature nuclei of murine rod photoreceptor cells exhibit a unique structure with a single large central chromocenter[64]. Most of the nuclei in the outer nuclear layer of young B6 mice exhibited this characteristic structure, whereas nuclei from $Rho^{P23H/+}$ and $Rho^{G188R/+}$ mice also exhibited nuclei with condensed chromatin (homogeneous nuclear staining) or disrupted structure (Fig. 8C, D), which are indicative of a dying cell. PROTEOSTAT appears to surround or coat the nuclei (Fig. 8D, E), which was observed for both visually normal and

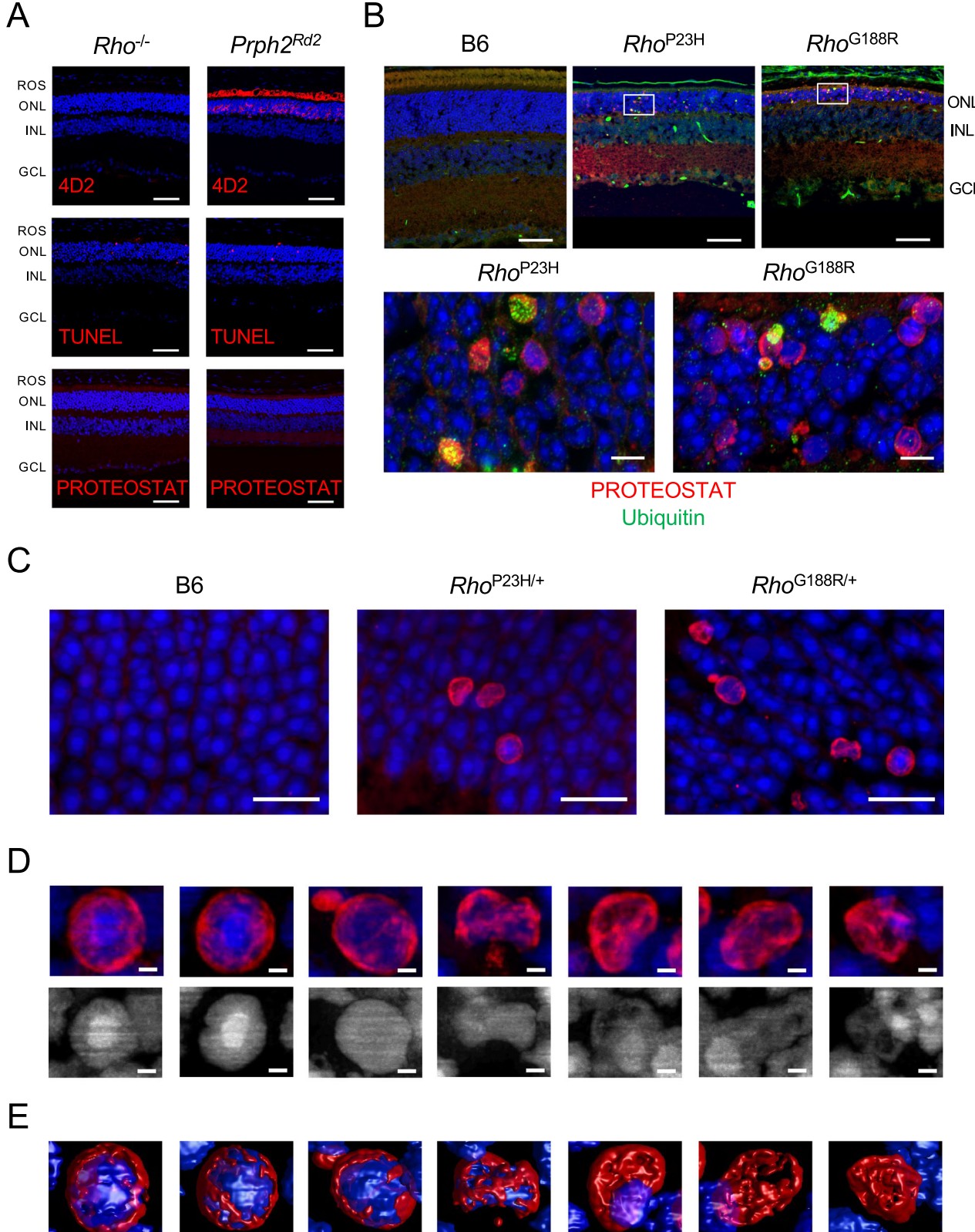

unhealthy nuclei. Similar morphology of PROTEOSTAT staining was also observed in *Rho*^P23H and *Rho*^G188R mice (Supplementary Fig. 2), where nuclei were generally unhealthier and there were more dying cells compared to those in heterozygous mutant mice. Large structures indicative of aggresomes, however, were not observed with PROTEOSTAT staining nor were they observed in EM images (Supplementary Fig. 3). Taken together, PROTEOSTAT staining in mice expressing misfolding mutants of rhodopsin appear to reflect the aggregation of misfolded rhodopsin mutants and is independent of retinal degeneration or rhodopsin mislocalization. Aggregates detected by PROTEOSTAT do not appear to form aggresomes.

**Fig. 8 | Characterization of PROTEOSTAT staining in the outer nuclear layer.**
**A** Retinal cryosections from 1-month-old *Rho*[-/-] mice and 2-month-old *Prph2*[Rd2] mice were labeled with the anti-4D2 antibody, TUNEL or PROTEOSTAT (red). Nuclei were labeled with DAPI (blue). Scale bar, 50 μm. The ROS, outer nuclear layer (ONL), inner nuclear layer (INL), and ganglion cell layer (GCL) are labeled. **B** Retinal cryosections from 2-week-old B6, *Rho*[P23H], and *Rho*[G188R] mice were colabeled with PROTEOSTAT (red) and an anti-ubiquitin antibody (green). Nuclei were labeled with NucBlue (blue). Lower magnification images (scale bar, 50 μm) are shown on top and higher magnification maximum intensity projection images (scale bar, 5 μm) in regions highlighted by white boxes in the lower magnification images are shown on the bottom. Separated individual images showing only a single label are presented in Supplementary Fig. 4. **C** High magnification maximum intensity projection images of PROTEOSTAT staining in the outer nuclear layer. Retinal cryosections from 2-week-old B6 and *Rho*[G188/+] mice and 3-week-old *Rho*[P23H/+] mice were stained with PROTEOSTAT (red), and nuclei were labeled with DAPI (blue). Scale bar, 10 μm. **D** Zoomed in images of individual nuclei in the outer nuclear layer from **C** are shown. The bottom row shows DAPI staining only in grayscale. Nuclei are ordered according to how healthy they appeared. Scale bar, 1 μm. Movies of 3D reconstructions of the individual nuclei are presented in Supplementary Movie 1, Supplementary Movie 2, Supplementary Movie 3, Supplementary Movie 4, Supplementary Movie 5, and Supplementary Movie 6. **E** 3D surface rendered images of nuclei presented in **D** labeled with DAPI (blue) and PROTEOSTAT (red). All images are representative of at least three different experiments.

## Discussion

Murine molecular and animal models were examined to better understand the effect of P23H and G188R mutations in rhodopsin that cause adRP. In vitro aggregation and localization profiles of murine P23H and G188R rhodopsin were similar to those reported previously for their human counterparts[33]. This similarity contrasts the different aggregation and localization profiles exhibited by bovine P23H rhodopsin in comparison to its human counterpart[34]. Thus, unlike bovine rhodopsin, murine rhodopsin mutants appear to be an adequate molecular model to examine molecular defects promoted by the P23H and G188R mutations in rhodopsin that cause adRP. Mouse models expressing murine forms of the P23H and G188R mutants may also serve as adequate animal models to examine the pathogenesis of adRP. The P23H rhodopsin knockin mouse examined here was previously shown to recapitulate many aspects of the human disease such as thinning of the outer nuclear layer, shortening of the ROS, rod function being affected to a greater extent than cone function, and exhibiting intraretinal gradient of degeneration with the inferior retina affected to a greater extent than the superior retina[20]. The G188R rhodopsin knockin mouse also shares many of these features. This mouse exhibits progressive photoreceptor cell loss with shortened ROS (Figs. 4A–F, 5E, F, and 6B), rod function is affected to a greater extent than cone function (Fig. 4G–I), and the inferior retina exhibits a faster rate of degeneration compared to that in the superior retina (Fig. 4E, F and Table 1). These effects are common regardless of the class of misfolding mutation harbored by rhodopsin. Thus, the G188R rhodopsin knockin mouse like the P23H rhodopsin knockin mouse is an adequate model to examine the pathogenesis of adRP caused by rhodopsin mutation.

Distinguishing features of the P23H and G188R rhodopsin mutants in in vitro studies are their aggregation and mislocalization (Figs. 2 and 3). The G188R mutant exhibits a more severe aggregation profile, which may be related to the proximity of the mutation to the bound chromophore 11-*cis* retinal[6] (Fig. 1B). Do these features play a role in the observed retinal degeneration in the mouse models? Although mutants besides the P23H rhodopsin mutant have mainly been characterized in heterologous expression systems, it is unclear to what extent in vitro cell culture studies mimic in vivo conditions. Previous characterization of the P23H rhodopsin knockin mouse failed to detect rhodopsin mislocalization using the anti-1D4 antibody, which led to the conclusion that mislocalization is not a feature of the mutant rhodopsin in vivo and questioned a role for mislocalization and aggregation in the pathogenesis of the disease[20]. In contrast to this observation, we demonstrate here that rhodopsin does indeed mislocalize in both the P23H and G188R rhodopsin knockin mice and that the apparent absence of mislocalization was due to a masking of the 1D4 epitope (Fig. 5E, F), which was suggested previously[65].

The propensity of misfolding rhodopsin mutants to aggregate in vivo is suggested by multiple lines of evidence. We previously demonstrated in vitro that misfolding mutants of rhodopsin, demonstrated to aggregate by FRET, migrate on Western blots not as monomers but only as higher molecular weight species corresponding to multiples of rhodopsin molecules[36]. The same pattern of bands, where the monomeric band is absent, is also observed here in Western blots of retinal extracts from both *Rho*[P23H] and *Rho*[G188R] mice (Fig. 5C), suggesting that both P23H and G188R rhodopsin mutants aggregate in vivo like they do in vitro. PROTEOSTAT staining in the outer nuclear layer of P23H rhodopsin and G188R rhodopsin knockin mice also indicates that the mutant rhodopsins aggregate. Control studies in *Rho*[-/-] mice demonstrate that PROTEOSTAT staining in the outer nuclear layer is not detecting a non-rhodopsin species generated as a byproduct of retinal degeneration (Fig. 8A). Control studies in *Prph2*[Rd2] mice demonstrate that retinal degeneration does not cause mislocalized WT rhodopsin to form a species, such as an aggregate, that can be stained by PROTEOSTAT (Fig. 8A). We have shown previously in vitro that although mislocalization of rhodopsin accompanies aggregation, the mislocalization in and of itself does not cause aggregation[66]. Taken together, PROTEOSTAT staining in the outer nuclear layer appears to derive from aggregated mutant rhodopsin.

Rhodopsin mutants can form a variety of types of aggregates in vitro ranging from small aggregates to larger aggresomes[30,31,36]. Although the suggested use of the PROTEOSTAT dye is for the detection of aggresomes, we do not see any evidence that aggresomes form in photoreceptor cells of mutant mice, nor were they observed previously in other animal models[23]. The staining by PROTEOSTAT instead appears to indicate that aggregates of rhodopsin coat the surface of photoreceptor cell nuclei (Fig. 8D, E). Aggregates of amyloidogenic proteins are also able to coat the surface of membranes in cells including nuclear membranes[67,68]. Both P23H and G188R rhodopsin mutants exhibit increased β-sheet structure in vitro[66], which is characteristic of amyloid-type aggregates, however, it is unclear the extent to which features of rhodopsin aggregates align with those of amyloid-type aggregates. Since PROTEOSTAT staining occurs in some apparently healthy nuclei with a single large central chromocenter (Fig. 8D), there is a possibility that the observed localization of apparent aggregates surrounding photoreceptor cell nuclei precedes and contributes to cell death. More work, however, will be required to test this idea. The toxic potential of rhodopsin aggregates has been demonstrated previously in heterologous expression systems where aggregates were shown to impair the ubiquitin-proteasome system[30,69]. Approaches that reduce rhodopsin aggregation in vitro appear to reduce retinal degeneration in mouse models[70–73], establishing a possible link between aggregation and retinal degeneration. We demonstrate here a correlation between levels of PROTEOSTAT and TUNEL positive cells (Fig. 7B, C), supporting the notion that rhodopsin aggregation can underlie the observed photoreceptor cell death.

Studies in animal models expressing P23H rhodopsin suggest that there may be alternate mechanisms leading to retinal degeneration besides those related to the mislocalization and aggregation of the mutant receptor. It has been suggested that a minor fraction of the P23H rhodopsin mutant can traffic to the ROS and exert toxic effects at the level of ROS discs[74]. The P23H rhodopsin mutant has been proposed to disrupt the proper formation of ROS discs and destabilize the

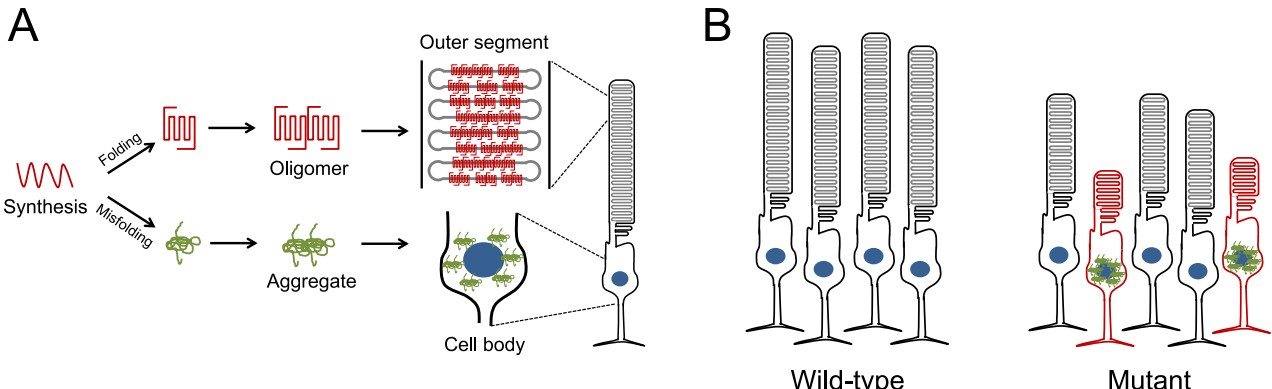

**Fig. 9 | Illustration of the relationship between rhodopsin aggregation and photoreceptor cell death. A** Properly folded rhodopsin adopts proper tertiary and quaternary structures and is transported to the ROS whereas misfolded rhodopsin is retained in the cell body of photoreceptor cells and can form aggregates. **B** In WT mice (left), healthy photoreceptor cells express a full complement of WT rhodopsin and form a long functional ROS. In mutant mice (right), photoreceptor cells expressing both WT and mutant rhodopsin will be functional but have a shorter ROS because most of the mutant is degraded and only half the complement of WT rhodopsin is expressed (black). Cell death occurs when photoreceptor cells stochastically form misfolded rhodopsin aggregates that surround the nucleus (red).

discs by forming aggregates within the disc membrane[21,74–76]. The degenerating retina in both *Rho*[P23H/+] and *Rho*[G188R/+] mice is accompanied by a shortened ROS (Figs. 5E, F and 6B), however, ROS discs are still able to form with rhodopsin packed into the membrane forming nanodomains that qualitatively resemble those previously characterized in B6 mice (Fig. 6A). This packing of rhodopsin in the membrane allows phototransduction to proceed as reflected in the scotopic a-wave of ERG traces (Fig. 4G). The presence of rhodopsin aggregates disrupting the normal nanodomain organization of rhodopsin was not evident, precluding this alternate explanation. Although no gross overall changes were observed in the packing of rhodopsin in ROS disc membranes, there were differences in the ROS disc properties compared to both B6 and *Rho*[+/-] mice (Fig. 6C–H). Perturbations in the morphogenesis of ROS discs have been noted previously in *Rho*[P23H/+] mice[21]. Differences in ROS disc properties also indicate that disc morphogenesis may be altered in mice expressing misfolding mutants of rhodopsin. It is unclear, however, whether this is caused directly by the mutants or is a byproduct of an unhealthy photoreceptor cell in a degenerating retina.

The kinetics of photoreceptor cell loss also suggest that the alternate mechanisms may not be a major factor in the observed retinal degeneration. Both mouse models exhibit similar trends in the rate of photoreceptor cell loss. Photoreceptor cell loss occurs at a faster rate in homozygous mutant mice compared to heterozygous mutant mice and the rate of photoreceptor cell loss in the inferior retina is greater than that of the superior retina, at least for heterozygotes (Table 1). The relative differences in the rate of photoreceptor cell loss are similar regardless of the point mutation in rhodopsin, thereby indicating that photoreceptor cell loss occurs by a common underlying mechanism in mice expressing both the P23H and G188R mutations. The alternate mechanisms discussed earlier are largely only applicable for the partial misfolding P23H rhodopsin mutant, which exhibits a less severe aggregation profile in vitro compared to the G188R mutant (Fig. 2). Consistent with these in vitro observations, a minor population of P23H mutant rhodopsin appears to traffic to the ROS and is signaling competent[21,22,75,77]. The more severe aggregation profile of the G188R rhodopsin mutant in vitro predicts that a similar minor population of mutant receptor that traffics and signals in the ROS does not exist in vivo, although this should be tested explicitly in the future. The major difference between mice expressing the P23H rhodopsin mutant or G188R rhodopsin mutant is a 2-fold difference in the rate of photoreceptor cell loss. Thus, the minor population of P23H rhodopsin that may traffic properly and is signaling competent appears to be beneficial rather than detrimental. The difference in

aggregation severity between the two mutants appears to underlie the 2-fold faster rate of photoreceptor cell loss observed in the G188R rhodopsin knockin mice.

The kinetics of photoreceptor cell loss and the progression of cell death and aggregation in mutant mice suggest a possible mechanism by which aggregation leads to photoreceptor cell loss. The exponential loss of photoreceptor cells in both mutant mouse lines is consistent with a one-hit model where photoreceptor cell loss occurs stochastically by a single event with either constant or decreasing variable cell death risk with age (Fig. 4E, F)[78,79]. The early peak and decline of TUNEL positive cells with age (Fig. 7B, C) is consistent with the latter[79]. The observed exponential kinetics of photoreceptor cell loss is inconsistent with mechanisms related to the cumulative damage hypothesis, where a gradual accumulation of toxic species occurs with age, resulting in increased risk of cell death with age and sigmoidal kinetics of photoreceptor cell death[80]. The progression of PROTEOSTAT positive cells mirrors that of TUNEL positive cells (Fig. 7B, C), indicating that the aggregates stained by PROTEOSTAT surrounding photoreceptor cell nuclei do not gradually form and accumulate over time, but rather, form stochastically in individual photoreceptor cells, perhaps due to a single nucleation event of aggregates[81]. The trigger for the stochastic event is unknown but may involve failure in some aspect of the proteostasis network including the chaperone and quality control system, proteasome activity or autophagy[82–85]. The implication of a stochastic photoreceptor cell death mechanism is that remaining photoreceptor cells can be functional and that therapeutics introduced at any time prior to the complete loss of photoreceptor cells can be beneficial[78]. The scotopic a-wave in ERG traces from heterozygous mutant mice is consistent with the notion that remaining photoreceptor cells are functional and capable of phototransduction (Fig. 4G). The stochastic nature of photoreceptor cell death is also consistent with therapeutic rescue experiments in other RP animal models where retinal degeneration could be halted in more advanced stages of retinal degeneration[86,87].

The overall picture emerging on the pathogenic effect of misfolding rhodopsin mutants based on studies here is summarized in Fig. 9A, B. Protein aggregation is a hallmark of many neurodegenerative diseases and rhodopsin aggregation appears to also belong in this category. Complete misfolding mutants like the G188R rhodopsin mutant appear to exclusively form aggregates and mislocalize whereas incomplete misfolding mutants like the P23H rhodopsin mutant can exist as a minor population of folded and properly trafficked receptor (Fig. 9A). Photoreceptor cells appear to be uniquely suited to degrade most of the misfolded mutant rhodopsin[20,22,88] (Fig. 5D). Thus, in most

photoreceptor cells, appreciable aggregation is not detected since the misfolded mutants are largely degraded. The removal of misfolded mutants results in lower levels of rhodopsin and a shorter ROS, which are still functional. Photoreceptor cells stochastically exhibit rhodopsin aggregates that coat the surface of their nuclei, which then leads to photoreceptor cell death (Fig. 9B). Based on this mechanism, interventions that can reduce rhodopsin aggregation are predicted to reduce the severity of retinal degeneration. Consistent with this view, the less severe aggregation profile of the P23H rhodopsin mutant results in a 2-fold slower rate of photoreceptor cell loss than that promoted by the G188R rhodopsin mutant (Table 1). Moreover, interventions that promote the proper folding of mutant rhodopsin (e.g., enhancing heat shock response or use of chaperones[73,89,90]) or enhance the clearance of misfolded rhodopsin mutants (e.g., increasing autophagy or proteasome activity[91–94]), have all shown beneficial effects in reducing retinal degeneration in mice expressing the P23H rhodopsin mutant. Thus, therapeutic strategies targeting the reduction or prevention of rhodopsin aggregation may be a viable option to combat adRP.

## Methods

### Mice
All animal studies reported here were conducted using protocols approved by the Institutional Animal Care and Use Committee at Case Western Reserve University School of Medicine. Mice were housed in rooms maintained at 22 °C and 50 % humidity under cyclic 12 h dark/12 h light. Mice were euthanized by carbon dioxide inhalation. Both male and female mice were used for experiments. No apparent sex differences were observed in preliminary assessments and therefore sex was not considered in the analyses of the data. $Rho^{P23H}$ (stock no. 017628), $Prph2^{Rd2}$ (stock no. 001979), and C57BL/6 J (stock no. 000664) mice were obtained from The Jackson Laboratory (Bar Harbor, ME). $Rho^{-/-}$ mice were obtained by backcrossing mice expressing a G90D rhodopsin mutant transgene on a null rhodopsin background (kindly provided by Dr. Paul Sieving, UC Davis, Sacramento, CA)[59,95] with C57BL/6J mice to remove the mutant transgene.

$Rho^{G188R}$ mice were generated using CRISPR/Cas gene targeting technology[96,97] at the Case Transgenic and Targeting Facility of Case Western Reserve University School of Medicine (Cleveland, OH). Fertilized embryos from C57Bl/6 J mice were injected with Cas9 nuclease (PNA Bio, Thousand Oaks, CA), sgRNA with the sequence 5'AGGGCATGCAATGTTCATGC (PNA Bio, Thousand Oaks, CA) and ssDNA replacement oligonucleotide with the sequence 5'TTTTATCATCCCTTGCGCTGACCATCAGGTACATCCCTGAGGGGATGCAATGTTCATGC**AGA**ATTGACTACTACACACTCAAGCCTGAGGTCAACAACGA (Integrated DNA Technologies, Coralville, IA), which contained the glycine (GGG) to arginine (AGA) mutation and ablates a protospacer adjacent motif site. Deep sequencing was conducted on the MiSeq System (Illumina, San Diego, CA) by the Genomics Core at Case Western Reserve University School of Medicine (Cleveland, OH) on samples from mosaic founder mice to identify mice with the desired mutation. A mouse harboring the mutation was identified and then mated with C57Bl/6 J mice. The progeny was genotyped to determine those exhibiting germline transmission of the point mutation. A 10,000 base pair region of the genome containing the rhodopsin gene and promoter region was sequenced by PCR-amplifying overlapping fragments to confirm that mice exhibited no changes except for the GGG to AGA mutation at codon 188. Mice were backcrossed with C57Bl/6 J mice for 5 generations to establish the line.

### In vitro studies in HEK293 cells
DNA constructs coding for WT, P23H, and G188R murine rhodopsin (mRho) tagged with a yellow fluorescent protein (YFP) variant containing a 1D4 epitope (pmRho-SYFP2-1D4, pmRhoP23H-SYFP2-1D4, and pmRhoG188R-SYFP2-1D4) were generated previously[66].

Constructs coding for murine rhodopsin and mutants tagged with mTurquoise2 (mTq2) containing a 1D4 epitope (pmRho-mTq2-1D4, pmRhoP23H-mTq2-1D4, and pmRhoG188R-mTq2-1D4) were generated by replacing human rhodopsin (hRho) in phRho-mTq2-1D4, generated previously[33], with WT or mutant murine rhodopsin at the EcoR1 and BamH1 restriction endonuclease sites.

Cells were transfected and prepared for the FRET assay or confocal microscopy as follows. HEK293T/17 cells (Cat. No. CRL-11268, American Type Culture Collection, Manassas, VA) were grown in Dulbecco's Modified Eagle's Medium - high glucose (Thermo Fisher Scientific, Waltham, MA), supplemented with 10% fetal bovine serum (Thermo Fisher Scientific, Waltham, MA) in 12-well plates and transiently transfected with DNA constructs described above (800 ng total) using Lipofectamine 2000 (Invitrogen, Carlsbad, CA)[6]. Cells for confocal microscopy were grown on poly-L-lysine treated #1.5 round coverslip glass (Thermo Fisher Scientific, Waltham, MA). Cells were either untreated or treated with 15 µM 9-*cis* retinal (MilliporeSigma, St. Louis, MO) 3 h after transfection under dim red-light conditions and incubated in the dark. Cells were assayed or imaged 24 h after transfection.

For the FRET assay, cells were washed and resuspended in 3 mL 1 × PBS (Thermo Fisher Scientific, Waltham, MA). The FRET assay was conducted on a FluoroMax-4 spectrofluorometer (Horiba Jobin Yvon, Edison, NJ). Fluorescence emission spectra of YFP (485 nm excitation, 5 nm slit width) and mTq2 (425 nm excitation, 5 nm slit width) were obtained from untreated cells, cells treated with 1.3 mM *n*-dodecyl-*β*-D-maltoside (DM) (Anatrace, Maumee, OH) for 5 minutes and then 3.3 mM SDS (Invitrogen, Carlsbad, CA) for 5 minutes. The FRET efficiency ($E$) was computed by measuring the dequenching of fluorescence from mTq2 at 476 nm[6]. Total FRET corresponds to the FRET signal from untreated cells and is composed of DM-sensitive and DM-insensitive FRET. DM-sensitive FRET is the FRET signal eliminated by treatment with DM corresponding to rhodopsin oligomers and DM-insensitive FRET is the FRET signal resistant to treatment with DM and corresponds to rhodopsin aggregates[36]. FRET curves were generated by plotting the FRET efficiency versus the acceptor:donor (A:D) ratio and fitting the data by non-linear regression to a rectangular hyperbolic function using Prism 9 (GraphPad Software, San Diego, CA): $E = (E_{max} \times A{:}D)/(EC_{50} + A{:}D)$. An extra sum of squares F test was conducted using Prism 9 (GraphPad Software, San Diego, CA) to compare each $E_{max}$ to the non-specific FRET $E_{max}$, which was defined previously[33].

Cells used for confocal microscopy were labeled with DAPI to stain the nuclei (Bio-Rad, Hercules, CA) and wheat germ agglutinin (WGA)-Alexa Fluor 647 conjugate (Invitrogen, Carlsbad, CA) to stain the plasma membrane, and the ER was labeled by cotransfecting cells with pDsRed2-ER (Takara Bio USA, Mountain View, CA)[66]. Confocal microscopy was performed on an SP8 confocal microscope (Leica, Buffalo Grove, IL) equipped with a 100x/1.4-NA oil objective[33]. DAPI was imaged by 405 nm diode laser excitation and 415 – 450 nm emission detection. mTq2 was imaged by 458 nm Argon laser excitation and 465 – 500 nm emission detection. YFP was imaged by 514 nm tunable white light laser excitation and 520 – 570 nm emission detection. DsRed2-ER was imaged by 558 nm tunable white laser excitation and 570 – 600 nm emission detection. WGA-Alexa Fluor 647 was imaged by 650 nm tunable white light laser excitation and 675 – 680 nm emission detection. Colocalization analysis of different fluorescent species in confocal microscopy images was conducted to compute the Pearson's correlation coefficient ($r$) using the Coloc 2 plugin in Fiji (version 2.1.0/1.53c)[34,98].

### Outer nuclear layer quantification
Mouse eyes were enucleated, processed, embedded, sectioned, and hematoxylin and eosin (H&E) stained by Excalibur Pathology (Norma, OK)[51]. H&E-stained sections were imaged on an Axio Scan.Z1 Slide

Scanner equipped with a Hitachi HV-F203 camera and a Plan Apo 20×/0.8-NA objective (Carl Zeiss Microscopy, White Plains, NY). The number of nuclei spanning the outer nuclear layer was counted manually on both the superior and inferior regions of the retina at various distances from the optic nerve. Three different sections from the same eye were quantified and averaged. To determine the kinetics of photoreceptor cell loss, values from 600, 800, and 1000 μm from the optic nerve were averaged and plotted. Data were fit with an equation for one-phase decay ($y = (y_0 - plateau) \times e^{-kx} + plateau$) using nonlinear regression to obtain the rate constant ($k$) using Prism 9 (GraphPad Software, San Diego, CA). The variable $y_0$ was set to be common among data sets analyzed together and the plateau was fixed to equal 1 to exclude the loss of cone photoreceptor cells.

### Electroretinography

ERG was conducted under dim red light conditions. Mice were dark adapted overnight and anaesthetized by intraperitoneal injection of a cocktail consisting of 20 mg/ml ketamine and 1.75 mg/ml xylazine (dose of 0.1 ml per 25 g body weight). Pupils were dilated with 1% tropicamide (Patterson Veterinary, Devens, MA) and the corneas were hydrated with 0.3% hypromellose (Alcon Laboratories, Fort Worth, TX). ERG was conducted on a Celeris rodent ERG system (Diagnosys, Lowell, MA) and analyzed with Espion 6.0 software (Diagnosys, Lowell, MA). Standard full-field stimulators with Ag/AgCl electrodes were placed on both eyes and the body temperature of mice was maintained throughout the recording session. ERG traces were obtained using touch/touch protocol and oscillatory potentials were filtered out. Scotopic responses were recorded with flash stimuli ranging from 0.001 to 20 cd·s/m². After 7 min of light adaptation at 20 cd·s/m², photopic responses were acquired with flash stimuli ranging from 0.13 to 63 cd·s/m². The a-wave and b-wave amplitudes obtained from traces from left and right eyes of the mouse were averaged. Data were plotted and fit by non-linear regression in Prism 9 (GraphPad Software, San Diego, CA) to a standard dose-response model, $R = \frac{R_{max}}{1 + 10^{\log K_A - \log I}}$, or biphasic dose-response model, $R = \frac{R_{max} \times f}{1 + 10^{\log K_A - \log I}} + \frac{R_{max} \times (1-f)}{1 + 10^{\log K_B - \log I}}$. $R$ is the amplitude of the a-wave or b-wave at a given flash intensity ($I$), $R_{max}$ is the maximal amplitude at a saturating flash intensity, $K_A$ and $K_B$ represents the flash intensity that generates a half-maximal amplitude, $f$ is the fraction of the curve that has $K_A$.

### Immunohistochemistry, TUNEL, and PROTEOSTAT assays

Retinal cryosections were prepared as follows. Eyes were fixed whole in 4% paraformaldehyde in 1 × PBS (Thermo Fisher Scientific, Waltham, MA) for 48 h at room temperature. Fixed eyes were cryoprotected in sucrose prepared in 1 × PBS: 15% sucrose for 12 h and then 30% sucrose overnight. Eyes were then embedded in OCT compound (Sakura Finetek. Torrance, CA), snap-frozen in liquid nitrogen and stored at -80 °C until sectioning. Frozen eyes were sectioned at 7 μm thickness on a Leica CM1950 cryostat (Leica Biosystems, Deer Park, IL). Some sections were prepared using the CryoJane Tape-Transfer System (Leica Biosystems, Deer Park, IL). Immunohistochemistry was conducted on cryosections using the following primary antibodies[99]: anti-1D4 (1:500 dilution)[46], anti-4D2 (1:1000 dilution, Cat. No. MABN15, MilliporeSigma, Burlington, MA), and anti-ubiquitin (1:100 dilution, Cat. No. sc-8017, Santa Cruz Biotechnology, Dallas, TX) antibodies. Primary antibodies were detected with a CF 647 goat anti-mouse secondary antibody (1:500 dilution, Cat. No. SAB4600183, MilliporeSigma, Burlington, MA). For antigen retrieval, sections were incubated in 10 mM Tris-HCl (pH 9) at 60 °C for 10 min and then stored at room temperature for 30 min. TUNEL assay was conducted on retinal cryosections using DeadEnd Fluorometric TUNEL System (Promega, Madison, WI), following the manufacturer's protocol. PROTEOSTAT staining of retinal cryosections was conducted using the PROTEOSTAT Aggresome Detection Kit (Enzo Life Sciences,

Farmingdale, NY), following the manufacturer's protocol. When immunohistochemistry was performed in parallel, immunohistochemistry was conducted after PROTEOSTAT staining was complete. Labeled cryosections were typically cover-slipped with DAPI Fluoromount-G mounting media (Southern Biotech, Birmingham, AL). For high magnification imaging, labeled cryosections were incubated with 0.5 μg/mL PUREBLU DAPI (Bio-Rad, Hercules, CA) for 10 min and then cover-slipped with ProLong Diamond Antifade Mountant (Invitrogen, Carlsbad, CA) or were cover-slipped with ProLong Glass Antifade Mountant with NucBlue stain (Invitrogen, Carlsbad, CA).

Confocal microscopy images were typically acquired on an Olympus FV1200 IX83 laser scanning confocal microscope equipped with a UPlanXApo 40 ×/1.40 NA oil objective (Evident/Olympus, Waltham, MA). High magnification images were acquired using a UPLXAPO 100 ×/1.45 NA objective with a 3 × digital zoom. Each image is a maximum intensity projection of up to 30 slices acquired with a 0.41 μm z-step size determined by the FV10-ASW 4.2 software (Evident/Olympus, Waltham, MA). Deconvolution was performed on z-stacks using the Autoquant plug-in (Media Cybernetics, Rockville, MD) within MetaMorph (version 7.7.8.0, Molecular Devices, San Jose, CA) or with cellSens Dimension 4.2 (Evident/Olympus, Waltham, MA) using constrained iterative algorithms. 3D reconstructions and movies were generated in MetaMorph from deconvolved z-stacks. 3D surface rendered images were generated with Huygens Professional 23.10 software (Scientific Volume Imaging, Hilversum, the Netherlands) using the Standard smart template in Deconvolution Express and the Surface Renderer. DAPI and NucBlue were imaged with 405 nm diode laser excitation and 425-460 nm emission detection. CF 647 was imaged with 635 nm diode laser excitation and 655-755 nm emission detection. TUNEL-positive cells that incorporated fluorescein-12-dUTP were detected by 473 nm argon-ion laser excitation and 485-545 nm emission detection. PROTEOSTAT dye was detected by 559 nm diode laser excitation and 575-620 nm emission detection. TUNEL, PROTEOSTAT, and DAPI positive cells were quantified in 317 × 317 μm images obtained at 700 – 1100 μm from the optic nerve on the superior and inferior regions of the retina. Quantification was performed in ImageJ (version 1.53n) by adjusting the threshold of the image and using the Analyze Particles function[100,101].

### Western blotting

Retinas were lysed in ice-cold RIPA buffer (Cell Signaling Technology; Danvers, MA) supplemented with a 1:100 dilution of a protease inhibitor cocktail (Cat. No. P8340, MilliporeSigma, Burlington, MA) and phenylmethylsulfonyl fluoride (MilliporeSigma, Burlington, MA), and stored at -80 °C. Lysates were prepared in Laemmli SDS Sample Buffer with reducing agent (Boston Bioproducts, Ashland, MA), loaded on a Novex 4-12% Tris-glycine gel (Invitrogen, Camarillo, CA) along with Precision Plus Protein Kaleidoscope Protein Standards (Bio-Rad, Hercules, CA), and SDS-PAGE performed. Western blotting was conducted using primary antibodies against rhodopsin (anti-1D4, 1:2500 dilution) and GAPDH (1:5000 dilution, Cat. No. 10494-1-AP; Proteintech, Rosemont, IL) and IRDye 800CW donkey anti-mouse (1:4000 dilution, Cat. No. 926-32212) or IRDye 680LT donkey anti-rabbit (1:4000 dilution, Cat. No. 925-68023) secondary antibodies (LI-COR Biosciences, Lincoln, NE). Western blots were imaged by the Odyssey Fc Imaging System (LI-COR Biosciences, Lincoln, NE). The intensity of all bands corresponding to rhodopsin were quantified on the LI-COR Image Studio 4.0 software (LI-COR Biosciences, Lincoln, NE). The summed intensities of rhodopsin bands were normalized to the intensity of the band corresponding to GAPDH.

### Quantitative real-time RT-PCR

Samples were prepared for quantitative real-time RT-PCR (RT-qPCR) as follows. Total RNA was isolated from retinal samples using High Pure RNA Tissue Kit (Roche Diagnostics, Indianapolis, IN) and reverse

transcription was performed using the Transcriptor First Strand cDNA Synthesis Kit (Roche Diagnostics, Indianapolis, IN). qPCR was conducted on the LightCycler 96 Real-Time PCR System (Roche Diagnostics, Indianapolis, IN). Primer sequences used for qPCR are those reported previously[102]: rhodopsin, forward (5′-CAAGAATCCACTGGGA GATGA), reverse (5′- GTGTGTGGGGACAGGAGACT); transducin, (5′-GAGGATGCTGAGAAGGATGC), reverse (5′-TGAATGTTGAGCGTGGT-CAT); 18s rRNA, forward (5′-TTTGTTGGTTTTCGGAACTGA), reverse (5′-CGTTTATGGTCGGAACTACGA). The relative levels of rhodopsin transcripts were normalized to that of 18S rRNA or transducin using the comparative $C_T$ method[103].

## Atomic force microscopy
All procedures were carried out under dim red light conditions. ROS disc membranes were prepared from the retinas of 13-16 dark-adapted mice that were 4 weeks of age[53], and were resuspended in Ringer's buffer (10 mM Hepes, 130 mM NaCl, 3.6 mM KCl, 2.4 mM $MgCl_2$, 1.2 mM $CaCl_2$, and 0.02 mM EDTA, pH 7.4). Contact mode AFM was performed on a Multimode II atomic force microscope equipped with an E scanner (Bruker, Santa Barbara, CA) using silicon nitride cantilevers with a nominal spring constant of 0.06 N/m (DNP-S, Bruker, Santa Barbara, CA). ROS disc membrane samples were adsorbed on freshly cleaved mica and imaged in 20 mM Tris, 150 mM KCl, 25 mM $MgCl_2$, pH 7.8[53]. Deflection images were analyzed using SPIP (version 6.7, Image Metrology A/S, Hørsholm, Denmark)[53]. Data were plotted using Prism 9 (GraphPad Software, San Diego, CA).

## Transmission electron microscopy
Mouse eyes were enucleated and fixed whole for 30 min in freshly made fixative (4% paraformaldehyde/2.5% glutaraldehyde in 0.2 M sodium cacodylate buffer, pH 7.4). The cornea and lens were then removed, and the sample was incubated in fixative overnight at 4 °C. Eye cups were washed with 0.2 M sodium cacodylate buffer, pH 7.4 three times for 5 min. Secondary fixation was carried out in 1% osmium tetroxide in water for 1 h at 4 °C. Eye cups were washed with 0.2 M sodium cacodylate buffer, pH 7.4 two times and then washed once with 0.05 M maleate buffer, pH 5.15 for 5 min. Eye cups were stained with 1% uranyl acetate in 0.05 M maleate buffer, pH 5.15 for 1 h and then washed with 0.05 M maleate buffer, pH 5.15 three times for 5 min. Samples were dehydrated in 30%, 50%, 75%, and 95% cold ethanol for 5 min and three times with 100% ethanol for 10 min. 100% ethanol was replaced with 100% ethanol/eponate 12 (Ted Pella Inc. Redding, CA) in a 1:1 ratio and incubated overnight at room temperature. Samples were then incubated in eponate 12 for 4–6 h at room temperature and then placed in a rubber mold and incubated for 24 h at 62 °C to promote polymerization. Ultra-thin sections of 85 nm were cut on a Leica EM UC7 Ultra-Microtome (Leica Microsystems, Buffalo, NY) with DiATOME Diamond Knives (Diatome, Hatfield, PA). Sections were stained with uranyl acetate and lead citrate for 7 min each and then imaged on a Tecnai G2 Spirit Bio electron microscope operated at 80 kV (FEI Company, Hillsboro, OR). Lower magnification images were taken at 2.8 kX and higher magnification images were taken at 30 kX magnification.

## Statistics
All statistical analyses were conducted using Prism 9 (GraphPad Software, San Diego, CA). Multiple comparisons were conducted using one-way ANOVA followed by post-hoc analysis to assess statistical significance of differences for individual comparisons. An extra sum of squares F test was conducted to determine statistical significance of parameters from fitted curves.

## Reporting summary
Further information on research design is available in the Nature Portfolio Reporting Summary linked to this article.

## Data availability
All data and images supporting the findings of this study are available within the paper and in Supplementary Information and Source Data files. Raw graphical data and uncropped Western blots are included in a Source Data file. Source data are provided with this paper.

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

## Acknowledgements

We thank Heather Butler for breeding and maintaining mouse colonies, John Denker for generating DNA constructs, genotyping mice, testing sgRNA for cutting efficiency and validating knockin mice, Dawn Smith for culturing HEK293 cells and generating cryosections, and Catherine

Doller for generating cryosections. We thank the Case Transgenic and Targeting Facility and the Genomics Core at Case Western Reserve University School of Medicine (Cleveland, OH) for generating and identifying G188R rhodopsin knockin mice. We would like to acknowledge the use of the Leica SP8 confocal microscope in the Light Microscopy Imaging Core at Case Western Reserve University made available through the Office of Research Infrastructure Programs (NIH-ORIP) Shared Instrumentation Grant S10 OD016164. We thank the Electron Microscopy Core at the Cleveland Clinic Lerner Research Institute (Cleveland, OH) and Mei Yin for preparing retinal sections and imaging by EM. This work was funded by grants from the National Institutes of Health (R01EY021731 (P.S.-H.P.), P30EY011373, and UL1RR024989), Ohio Lions Eye Research Foundation (P.S.-H.P.), and Cleveland Eye Bank Foundation (P.S.-H.P.).

## Author contributions

S.V. conducted experiments. S.V. and P.S.-H.P. designed experiments and analyzed data. S.S. conducted AFM studies. M.P. conducted some of the confocal microscopy studies. P.S.-H.P. wrote the manuscript. All authors reviewed and edited the manuscript.

## Competing interests

The authors declare no competing interests.
