## [Peer Review File · Nature Communications]

REVIEWER COMMENTS

Reviewer #1 (Remarks to the Author):

Progressive aggregation of rhodopsin mutants in mouse models of autosomal dominant retinitis pigmentosa

Vasudevan et al

This study by Paul Park and colleagues is a detailed and thorough investigation of a new mouse model of rhodopsin retinitis pigmentosa (RP) and comparison to the well characterized P23H model. Rhodopsin RP is one of the most common causes of RP and is most often associated with a single misfolding mutation causing dominant disease. The archetypal misfolding mutant P23H can be pharmacologically rescued. Here the authors have compared P23H to another mutant G188R which has a more severe misfolding phenotype. They have produced a new knock-in mouse model and performed extensive characterization. The model shows faster degeneration than P23H and more disrupted rod outer segments (ROS). The ROS were studied by EM and AFM. They identify inclusions of aggregated protein using the proteostat stain and correlate the presence of inclusions with photoreceptor cell death. The study is well performed, extremely detailed and clearly described. The following should be addressed.

Major.

1. The imaging of proteostat staining with ubiquitin and vimentin is too low magnification and the images small (even in the inset panels) and it is hard to see any co-labelling with the dual label. Split channels of higher resolution images should be shown so that it can be more clearly seen if there is co-localization of the ubiquitin and proteostat, and if the vimentin cytoskeleton has collapsed around the inclusion. Based on these images these inclusions are very large, almost the size of the photoreceptor nuclei. This seems unlikely and especially if they are meant to proceed cell death. In other neurodegenerative diseases large inclusions have been suggested to be protective, whereas smaller aggregates might be the most toxic species.
2. Ideally, the authors would also provide some orthogonal evidence of aggregation in vivo. For example, it has previously been shown that P23H aggregation can be monitored by differential sedimentation and reduces sedimentation correlates with photoreceptor survival. Alternatively, could these inclusions as electron dense material be observed in the TEM?
3. If, as the authors suggest, no G188R traffics to the ROS then why is the yield and structure of the ROS in the G188R/+ so severely affected? If there is no effect of G188R on the WT rhodopsin and the protein mutant does not traffic to the ROS, surely the ROS should resemble the Rho+/- ROS with reduced rhodopsin only?

Minor.

1. Abstract: the authors claim that 'A distinguishing feature of these mutants in vitro is that they mislocalize and aggregate. It is unclear whether these features occur in vivo and play a prominent role in causing retinal degeneration.' Mislocalization has been well studied both in vitro and in vivo for P23H (and other types of rhodopsin mutants), through immunohistochemistry, immunoEM and biochemical assessment. Furthermore, whilst it is novel that they have observed inclusions in the ONL of the G188R model, other reports have described proteostat positive staining in the P23H and D190N models, as well as inclusions in other models of rhodopsin RP (e.g. in *Xenopus*) and shown aggregated sedimentable rhodopsin in retina (in P23H rats). This statement should be clarified or modified.

2. Introduction: 'Moreover, variable observations have been made in different animal models expressing the P23H rhodopsin mutant (20-25), presenting challenges in assessing pathogenic mechanisms.' The major differences reported here refer to either gene dosage (transgenic overexpression versus knock-in at endogenous levels), or the presence of a GFP tag, and I feel the authors are overstating any differences as the consensus is that the P23H protein is predominantly ER retained and degraded in these models.

3. Figure 4 and Table 1. The authors should consider how cone survival might affect their estimates of the rate of degeneration. It is likely that the single row of nuclei observed in the models at some time points are surviving cone cells (e.g. homozygous mice at 1 month or inferior retina in G188R/+) For example, this row has very low levels of rhodopsin immunoreactivity consistent with them being cones. The mechanisms of this non-cell autonomous cone cell death (moving from one row to 0) are likely to be different from rhodopsin mediated rod cell death. Also once 1 row remains the rod cells have probably all been lost already leading to an underestimate of the effect on rods.

4. Figure 5. The authors speculate that 'Thus, the masking of the 1D4 epitope in the mutant mice may indicate that those rhodopsin molecules are aggregated, as is predicted to occur from in vitro studies.' Do they have any evidence that the 1D4 epitope is masked in their in vitro studies? 1D4 can be affected by phosphorylation and epitopes are more generally masked by fixation so there are other potential explanations and this is very indirect as an assessment of aggregation.

5. Figure 6. It would be useful to show the B6 ROS AFM images for direct comparison rather than referring to published data. It would also be better if there was quantification of the disc membranes displaying a discernible rim region and rhodopsin nanodomains in the lamellar region in all three models rather than 'infrequently observed' in the RhoG188R/+, especially given that the yields were not sufficient for the detailed image analyses.

6. Proteostat. Proteostat is a commercially available dye. It does not specifically recognize 'aggresomes' (as described by the manufacturer), rather it is designed to intercalate into the cross-beta spine of quaternary protein structures typically found in misfolded and aggregated proteins (collapsed beta sheets characteristic of amyloid). By contrast aggresomes, are a cellular inclusion of aggregated proteins dependent on microtubules as the authors describe. The should revise the description of what proteostat binds.

7. Why does the proteostat stain the outer segments of B6 mice (as well as control), could it be that it binds rhodopsin? Also why is the background intensity of the staining greater between the INL and GCL in the G188R models?

Reviewer #2 (Remarks to the Author):

What are the noteworthy results?

The authors show that P23H rhodopsin (the mutation most commonly responsible for autosomal dominant RP in humans, and the subject of many previous in vitro and in vivo studies) is mislocalized and forms aggregates in mouse photoreceptors. In addition, G188R rhodopsin, (a different mutation that is more prone to aggregate, and not previously studied in vivo) is also mislocalized and forms aggregates in mouse photoreceptors; it is more prone to forming these aggregates, and causes more aggressive retinal degeneration. The aggregates have properties of aggresomes, a specific type of intracellular inclusion associated with microfilaments and ubiquitination. Properties of the outer segment disks are also reported: P23H^{+/-} and G188R^{+/-} disks are more disordered by EM and AFM. P23H also have smaller disk nanodomains apparent by AFM than either WT or rho^{+/-} disks; G188R had too few disks to provide AFM measurements. As mislocalization of P23H rhodopsin and electron microscopy showing disordered disks have been reported previously, the chief novel findings are related to G188R, the presence of intracellular aggregates, and the AFM results for P23H^{+/-} disks. However, the in-vivo data on aggregates is somewhat problematic (see below). In vitro studies of G188R and P23H rhodopsins aggregating using FRET are also reported; however, this lab published similar experiments in 2016 and 2018 using the same rhodopsin mutants using a human rhodopsin cDNA; here the studies are replicated with mouse rhodopsins, and similar results were obtained.

Will the work be of significance to the field and related fields? How does it compare to the established literature? If the work is not original, please provide relevant references.

The work adds to the field and would likely be cited by authors in the field. However, it is not clear that this paper represents a major advance; the data is definitely interesting, but somewhat confirmatory or iterative in its importance.

Does the work support the conclusions and claims, or is additional evidence needed?

The in-vivo evidence of aggregates that resemble aggresomes is problematic.

1) The fluorescence microscopy shows aggregates at too low magnification; they appear to be brightly labeled cell bodies, not subcellular structures. They appear at the level of the outer nuclear layer (i.e. photoreceptor nuclei), which is not consistent with the original description of aggresomes as pericentriolar structures (which would be expected in the inner segment).

2) Although electron microscopy was carried out by the authors for other purposes (i.e. imaging the organization of ROS disks), there is no attempt by the authors to provide evidence of aggresome-like structures at the EM level. Previous studies with extensive electron microscopy on the P23H knock-in animals also did not report aggresome-like structures. (Sakami et al 2014, Sakami et al 2011), but this was a similar scenario of non-reporting, not negative findings.

3) In Fig 7, the PROTEOSTAT dye used by the authors strongly labels structures that have not been demonstrated to contain aggresomes (i.e. WT rod outer segments) with no explanation provided by the authors, suggesting that this dye does not exclusively label aggresomes, and/or labels non-aggregated rhodopsin.

4) Similarly, ubiquitin labeling is unexpectedly (to me?) present in OS of WT animals, without explanation.

5) Co-localization with ubiquitin and vimentin labeling with PROTEOSTAT labeling is used as evidence for the presence of aggresomes. However, ubiquitin and vimentin labeling is uniformly brighter in all retinal cell types in knock-in animals of both types, not just rod photoreceptors as would be expected, with no explanation provided by the authors.

6) Previous reports indicate vimentin is only expressed in a subset of retinal cells, which does not include photoreceptors. Generally, vimentin is not found in mature neurons; instead, neurofilaments are present.

Are there any flaws in the data analysis, interpretation and conclusions? - Do these prohibit publication or require revision?

Is the methodology sound? Does the work meet the expected standards in your field?

The author is a noted expert on atomic force microscopy of photoreceptors and FRET spectroscopy, with several high-profile publications. These aspects of the work seem well done and are likely high quality, though I am not an expert on either technique. The more standard forms of analysis also seem well done, with the exceptions noted above. In my opinion, the article cannot claim to have identified aggregates/aggresomes in vivo without improving the fluorescence study, providing electron microscopic evidence, and providing western blots from P23H/P23H and G188R/G188R retinas. Currently, the only evidence of in-vivo aggregation is from dye labeling, which also labels outer segments of WT retina containing (presumably) un-aggregated rhodopsin. It is not critical that the mutant aggregated rhodopsin is found in a structure that can be termed an "aggresome". If the structures differ from canonical aggresomes in notable ways, and have not been previously reported, that would also be interesting.

Is there enough detail provided in the methods for the work to be reproduced?

Yes

Detailed points:

Quote from the abstract: "A distinguishing feature of these mutants in vitro is that they mislocalize and aggregate. It is unclear whether these features occur in vivo and play a prominent role in causing retinal degeneration."

This is misleading. Although it has not, to my knowledge, been demonstrated in humans, multiple studies using multiple different animal models of P23H rhodopsin have demonstrated that P23H rhodopsin mislocalizes. The authors note later in the manuscript that the initial study of the P23H knock-in mouse did not identify mis-localization; however, other examples have been reported in other transgenic animals.

Fig 5B (western blot): there is no evidence of aggregation in this western blot. Is it possible to determine what fraction of the protein is P23H rhodopsin? Much of the rhodopsin signal in this blot is likely WT

rhodopsin from OS. What would a blot of P23H/P23H or G188R/G188R retina look like? In this scenario, any aggregation should be much more obvious.

Fig 6: No AFM images of WT disks are included. Although this has been previously published, this control would be useful to include to allow the reader to conduct a comparison. Are aggresomes visible in the authors' electron microscopy images? The original descriptions of aggresomes (Johnston and Kopito) included electron microscopy images showing a dense pericentriolar structure surrounded by a cage of filaments. The authors provide no investigation of the inner segments or cell bodies to determine the nature of the structures labeled in the fluorescence study.

Fig 7C: please comment on the fact that the PROTEOSTAT dye and anti-ubiquitin seem to unexpectedly intensely label rod outer segments in WT mice; what is the evidence that PROTEOSTAT specifically labels aggregated rhodopsin as opposed to all rhodopsin? Please comment on the much higher levels of ubiquitin and vimentin staining in all retinal layers and cell types in the mutant mouse retina. Please comment on the fact that vimentin has been reported to be expressed only in developing neurons, whereas mature neurons express neurofilaments. In the retina, vimentin has been reported to be expressed in Muller cells, astrocytes, and horizontal cells. However, the labeling of WT retina with anti-vimentin in Figure 7C does not appear to be restricted to specific cell types. What is the evidence that the labeled structures are aggresomes and not (e.g.) autophagosomes? A possibility is that the labeled cells are undergoing a proteostatic crisis in which the labeled structures are autophagosomes. This would be consistent with previous publications (Yao and Zacks 2018, Yao and Zacks 2016, Boga et al 2015, Wen et al., 2019)

Figure 8: Why are no results presented for WT mice?

Figure 8 C, D, and discussion: I suggest including more discussion regarding how this model differs significantly from that of Arshavsky and Lobanova, in which it is proposed that retinal degeneration in P23H rhodopsin photoreceptors and other forms of retinal degeneration occurs when the machinery for dealing with misfolded proteins (the proteasome) becomes saturated ("proteostatic crisis")? (Lobanova et al 2013 and other papers by Lobanova/Arshavsky). Are there novel predictions of this model?

Quote: "Smaller aggregates of misfolded rhodopsin mutants are also likely present in the retina of the mutant mice investigated here as well, although we are currently unable gain access to this information." This is unusual phrasing. What does this mean? Techniques do not exist, or are not accessible to the authors, or that the information already exists somewhere that is inaccessible to the authors?

Manuscript NCOMMS-23-15269

“Aggregation of rhodopsin mutants in mouse models of autosomal dominant retinitis pigmentosa”

We thank both reviewers for their valuable comments and suggestions on our manuscript. We have considered each comment and in response have conducted additional experiments and modified the text of the manuscript accordingly. We believe these changes have strengthened the manuscript further. Provided below are our responses to specific comments made by the reviewers.

Response to Reviewer #1

Reviewer #1:

This study by Paul Park and colleagues is a detailed and thorough investigation of a new mouse model of rhodopsin retinitis pigmentosa (RP) and comparison to the well characterized P23H model. Rhodopsin RP is one of the most common causes of RP and is most often associated with a single misfolding mutation causing dominant disease. The archetypal misfolding mutant P23H can be pharmacologically rescued. Here the authors have compared P23H to another mutant G188R which has a more severe misfolding phenotype. They have produced a new knock-in mouse model and performed extensive characterization. The model shows faster degeneration than P23H and more disrupted rod outer segments (ROS). The ROS were studied by EM and AFM. They identify inclusions of aggregated protein using the proteostat stain and correlate the presence of inclusions with photoreceptor cell death. The study is well performed, extremely detailed and clearly described. The following should be addressed.

Response:

We thank the reviewer for noting our studies were performed well, extremely detailed, thorough, and clearly described. We hope that we have addressed all concerns raised by the reviewer below.

Major Issues

Reviewer #1:

1. The imaging of proteostat staining with ubiquitin and vimentin is too low magnification and the images small (even in the inset panels) and it is hard to see any co-labelling with the dual label. Split channels of higher resolution images should be shown so that it can be more clearly seen if there is co-localization of the ubiquitin and proteostat, and if the vimentin cytoskeleton has collapsed around the inclusion. Based on these images these inclusions are very large, almost the size of the photoreceptor nuclei. This seems unlikely and especially if they are meant to proceed cell death. In other neurodegenerative diseases large inclusions have been suggested to be protective, whereas smaller aggregates might be the most toxic species.

Response:

Our primary goal in the original manuscript was to determine whether rhodopsin aggregation plays a role in photoreceptor cell death in adRP. Our data revealing a correlation between

aggregation and photoreceptor cell death demonstrate that aggregation is important in adRP, and we hope these results promote a renewed effort to better understand how aggregation causes photoreceptor cell death and determine which aggregate species are responsible. Moreover, the kinetics of cell death and detection of aggregates by PROTEOSTAT suggests a stochastic mechanism for retinal degeneration rather than one involving the gradual accumulation of a toxic species. We have added this mechanistic insight in the Discussion of the revised manuscript.

Our attempt to classify aggregates stained by PROTEOSTAT was a secondary issue in the paper. We based our classification on the advertised use of the PROTEOSTAT kit to detect aggresomes, and some apparent colocalization with markers in low magnification images. The reviewer is correct in pointing out that low magnification images can be misleading. In response to the reviewer's concerns, we conducted high magnification confocal microscopy, and additionally collected multiple z-stack images to obtain 3D information, including the generation of 3D movies (Figs. 8B-8D). These new data reveal a distinct morphology for PROTEOSTAT staining that was not apparent from low magnification images. The morphology of PROTEOSTAT staining is not consistent with aggregates forming large inclusions such as aggresomes, but rather, aggregates appear to coat the surface of photoreceptor cell nuclei, a behavior that is also observed with some amyloid-type aggregates. New electron microscopy studies also demonstrate the absence of large inclusions indicative of aggresomes. Thus, aggregates detected in mutant mice do not appear to form aggresomes, as we first presumed based on the description of the PROTEOSTAT kit. This type of morphology of aggregates observed in our mutant mice is a new finding and we thank the reviewer for the suggested experiments.

The original anti-ubiquitin antibody used in the manuscript was the same one used in the original characterization of PROTEOSTAT as a dye to detect aggresomes¹, where colocalization of ubiquitin and PROTEOSTAT was demonstrated. Unfortunately, this antibody was discontinued, and we used a new anti-ubiquitin antibody from Santa Cruz to repeat experiments. There were some differences in staining patterns, possibly due to different specificities of the two antibodies. Similar to the original antibody, the new anti-ubiquitin antibody showed staining in the outer nuclear layer of mutant mice with some colocalization with PROTEOSTAT. High magnification imaging, however, indicated that the structures colabeled with PROTEOSTAT and anti-ubiquitin antibody do not represent aggresomes and are not associated with the nucleus. Similar patterns were observed with 2 other antibodies we tested. This new data is presented in Fig. 8B of the revised manuscript.

We have removed data with anti-vimentin antibody staining since our new characterizations revealed that aggresomes do not form, and therefore these data are no longer necessary. We did conduct higher magnification imaging with anti-vimentin antibody staining and also tested other antibodies in addition to our original antibody, which was used previously to study aggresomes^{2,3}. The high magnification imaging revealed that vimentin is present in the outer nuclear layer, but likely represents Müller cells engulfing dead photoreceptor cells rather than aggresomes.

Reviewer #1:

2. Ideally, the authors would also provide some orthogonal evidence of aggregation *in vivo*. For example, it has previously been shown that P23H aggregation can be monitored by differential sedimentation and reduces sedimentation correlates with photoreceptor survival. Alternatively, could these inclusions as electron dense material be observed in the TEM?

Response:

We provide additional evidence that the mutants aggregate in the retina of mutant mice by Western blot. We show in Western blots that rhodopsin in the mutant mice exhibit a different pattern of bands than wild-type rhodopsin. Wild-type rhodopsin predominantly migrates as a monomer whereas the mutants migrate predominantly as complexes of two or more rhodopsin molecules (Fig. 5C). This pattern in Western blots is the same as we observed previously for heterologously expressed misfolding rhodopsin mutants where we directly detected aggregates by FRET⁴. This additional data further supports the idea that the mutants form aggregates *in vivo*. Our new electron microscopy images demonstrate that aggregates do not form large aggregosomal structures and our new confocal microscopy images reveal aggregates coating the surface of photoreceptor cell nuclei.

Reviewer #1:

3. If, as the authors suggest, no G188R traffics to the ROS then why is the yield and structure of the ROS in the G188R/+ so severely affected? If there is no effect of G188R on the WT rhodopsin and the protein mutant does not traffic to the ROS, surely the ROS should resemble the Rho^{+/-} ROS with reduced rhodopsin only?

Response:

The ROS of *Rho*^{P23H/+} and *Rho*^{G188R/+} mice should resemble those of *Rho*^{+/-} mice in the absence of any toxic effects from the misfolding rhodopsin mutants. The misfolding mutants, however, form aggregates that appear to be toxic and cause retinal degeneration. Thus, *Rho*^{P23H/+} and *Rho*^{G188R/+} mice and *Rho*^{+/-} mice are not equivalent, since the latter does not exhibit retinal degeneration. In both *Rho*^{P23H/+} and *Rho*^{G188R/+} mice, there are clearly perturbations to ROS structure. Greater perturbations are observed for *Rho*^{G188R/+} mice, where retinal degeneration is more severe, suggesting a byproduct of retinal degeneration are perturbations to disc morphogenesis. We had suggested this in the original manuscript and include an additional sentence in the Discussion to reiterate this idea.

Minor Issues**Reviewer #1:**

1. Abstract: the authors claim that ‘A distinguishing feature of these mutants *in vitro* is that they mislocalize and aggregate. It is unclear whether these features occur *in vivo* and play a prominent role in causing retinal degeneration.’ Mislocalization has been well studied both *in vitro* and *in vivo* for P23H (and other types of rhodopsin mutants), through immunohistochemistry, immunoEM and biochemical assessment. Furthermore, whilst it is novel that they have observed inclusions in the ONL of the G188R model, other reports have described

proteostat positive staining in the P23H and D190N models, as well as inclusions in other models of rhodopsin RP (e.g. in *Xenopus*) and shown aggregated sedimentable rhodopsin in retina (in P23H rats). This statement should be clarified or modified.

Response:

We did not intend to disregard these studies, many that we referenced in our manuscript, in the sentence highlighted in the abstract. We have modified this sentence to now read, “It has been questioned whether these features occur *in vivo* and it is unclear whether they play a prominent role in causing retinal degeneration.”

Reviewer #1:

2. Introduction: ‘Moreover, variable observations have been made in different animal models expressing the P23H rhodopsin mutant (20-25), presenting challenges in assessing pathogenic mechanisms.’ The major differences reported here refer to either gene dosage (transgenic overexpression versus knock-in at endogenous levels), or the presence of a GFP tag, and I feel the authors are overstating any differences as the consensus is that the P23H protein is predominantly ER retained and degraded in these models.

Response:

We have modified this sentence to better highlight that differences are observed depending on the type of animal model generated. It now reads, “Moreover, differences are observed depending on the type of animal model generated expressing the P23H rhodopsin mutant, presenting challenges in assessing pathogenic mechanisms.”

Reviewer #1:

3. Figure 4 and Table 1. The authors should consider how cone survival might affect their estimates of the rate of degeneration. It is likely that the single row of nuclei observed in the models at some time points are surviving cone cells (e.g. homozygous mice at 1 month or inferior retina in G188R/+) For example, this row has very low levels of rhodopsin immunoreactivity consistent with them being cones. The mechanisms of this non-cell autonomous cone cell death (moving from one row to 0) are likely to be different from rhodopsin mediated rod cell death. Also once 1 row remains the rod cells have probably all been lost already leading to an underestimate of the effect on rods.

Response:

To exclude the effect of cone photoreceptor loss, we reanalyzed the kinetic data so that the plateau was fixed to equal 1, assuming a single row of nuclei represents cone photoreceptor cells. Minor changes to the fitted value for the rate constant, k , were observed, however, all the relative differences were maintained and did not change the conclusions from the data. We include a description of this change in fitting the kinetic data in the Materials and Methods and updated Table 1 and Fig. 4.

Reviewer #1:

4. Figure 5. The authors speculate that ‘Thus, the masking of the 1D4 epitope in the mutant mice may indicate that those rhodopsin molecules are aggregated, as is predicted to occur from in vitro studies.’ Do they have any evidence that the 1D4 epitope is masked in their in vitro studies? 1D4 can be affected by phosphorylation and epitopes are more generally masked by fixation so there are other potential explanations and this is very indirect as an assessment of aggregation.

Response:

We do not have direct evidence that the masking of the 1D4 epitope is related to aggregation. We have modified this sentence to reflect that this is one possible explanation. The sentence now reads, “Thus, one explanation for the masking of the 1D4 epitope in the mutant mice may be related to rhodopsin aggregation.”

Reviewer #1:

5. Figure 6. It would be useful to show the B6 ROS AFM images for direct comparison rather than referring to published data. It would also be better if there was quantification of the disc membranes displaying a discernible rim region and rhodopsin nanodomains in the lamellar region in all three models rather than ‘infrequently observed’ in the *RhoG188R/+*, especially given that the yields were not sufficient for the detailed image analyses.

Response:

We did not present images of ROS disc membranes from B6 mice since we had published several papers reporting similar images. We include in the revised manuscript a couple sample images to give readers a reference image without the need to search our past publications. Due to heterogeneity in ROS discs, it is difficult to discern differences in disc membrane properties unless differences are rather dramatic, which is why we need to conduct quantitative analysis.

It is difficult to quantify the number of proper discs obtained from *Rho*^{G188R/+} mice in a manner where we could compare “yields” with samples from the other mice. What we present are membranes that we were able to observe over several days of AFM imaging. Even for wild-type mice, there is batch to batch variability in the amount of disc membranes observed and the time it takes to detect and image discs. We currently do not have a way to report the yield of discs in a standardized way that would be informative.

Reviewer #1:

6. Proteostat. Proteostat is a commercially available dye. It does not specifically recognize ‘aggresomes’ (as described by the manufacturer), rather it is designed to intercalate into the cross-beta spine of quaternary protein structures typically found in misfolded and aggregated proteins (collapsed beta sheets characteristic of amyloid). By contrast aggresomes, are a cellular inclusion of aggregated proteins dependent on microtubules as the authors describe. The should revise the description of what proteostat binds.

Response:

The reviewer is correct in that PROTEOSTAT binds protein aggregates and is not necessarily specific for aggresomes, which is now demonstrated with our new data. We have modified the text to specifically indicate that PROTEOSTAT binds aggregated proteins and becomes fluorescent. The text now reads, “To detect rhodopsin aggregates in the retina, retinal cryosections were stained with PROTEOSTAT, which is a molecular rotor dye that becomes fluorescent upon binding to aggregated proteins.”

Reviewer #1:

7. Why does the proteostat stain the outer segments of B6 mice (as well as control), could it be that it binds rhodopsin? Also why is the background intensity of the staining greater between the INL and GCL in the G188R models?

Response:

We focused on PROTEOSTAT staining in the outer nuclear layer, which was distinct from staining in other regions of the retina, including the outer segments, that were fainter and variable in intensity and may represent non-specific staining. Evidence that the staining in the outer nuclear layer represents rhodopsin aggregates is shown in control studies demonstrating that neither photoreceptor cell death nor mislocalization of rhodopsin results in PROTEOSTAT-positive signals in the outer nuclear layer (Fig. 8A). We provide a more explicit description in the revised manuscript and indicate that staining outside of the outer nuclear layer may be non-specific.

Response to Reviewer #2**Reviewer #2:**

The authors show that P23H rhodopsin (the mutation most commonly responsible for autosomal dominant RP in humans, and the subject of many previous in vitro and in vivo studies) is mislocalized and forms aggregates in mouse photoreceptors. In addition, G188R rhodopsin, (a different mutation that is more prone to aggregate, and not previously studied in vivo) is also mislocalized and forms aggregates in mouse photoreceptors; it is more prone to forming these aggregates, and causes more aggressive retinal degeneration. The aggregates have properties of aggresomes, a specific type of intracellular inclusion associated with microfilaments and ubiquitination. Properties of the outer segment disks are also reported: P23H^{+/-} and G188R^{+/-} disks are more disordered by EM and AFM. P23H also have smaller disk nanodomains apparent by AFM than either WT or rho^{+/-} disks; G188R had too few disks to provide AFM measurements. As mislocalization of P23H rhodopsin and electron microscopy showing disordered disks have been reported previously, the chief novel findings are related to G188R, the presence of intracellular aggregates, and the AFM results for P23H^{+/-} disks. However, the in-vivo data on aggregates is somewhat problematic (see below). In vitro studies of G188R and P23H rhodopsins aggregating using FRET are also reported; however, this lab published similar experiments in 2016 and 2018 using the same rhodopsin mutants using a human rhodopsin cDNA; here the studies are replicated with mouse rhodopsins, and similar results were obtained.

The work adds to the field and would likely be cited by authors in the field. However, it is not clear that this paper represents a major advance; the data is definitely interesting, but somewhat confirmatory or iterative in its importance.

Response:

We thank the reviewer for acknowledging that our work will be a useful contribution to the field and that our data will be of interest. We also thank the reviewer for their comments and suggestions, which has improved the manuscript and provided even more insights about the aggregation of rhodopsin mutants. We have revised the manuscript to address the reviewer's comments and more explicitly stated what our data suggests in terms of mechanism of retinal degeneration. We believe our original and revised data firmly support the importance of aggregation in adRP caused by misfolding rhodopsin mutants and that it will spur future studies to better characterize, both structurally and functionally, aggregates formed by rhodopsin mutants. In addition to highlighting the importance of aggregation, our longitudinal studies provide mechanistic insight into how photoreceptor cell death occurs in mice expressing misfolding rhodopsin mutants that we failed to highlight in the original manuscript. As described in more detail below, our data indicate photoreceptor cell loss occurs stochastically. Longitudinal studies are often not conducted examining various potential factors that contribute to retinal degeneration, but they can provide novel insights.

Reviewer #2:

The in-vivo evidence of aggregates that resemble aggresomes is problematic.

Response:

The assignment of aggregates as aggresomes was secondary to the primary findings that misfolding rhodopsin mutants aggregate (shown *in vitro* and *in vivo*) and that the progression of aggregation mirrors photoreceptor cell death as monitored by TUNEL, which demonstrates that aggregation plays a prominent role in retinal degeneration in adRP. We attempted to give a first approximation of the type of aggregates formed by misfolding mutants of rhodopsin based on the description of the commercially available PROTEOSTAT reagent as a detection kit for aggresomes and apparent colocalization with markers in low magnification images. Based on the reviewer's comments, we have conducted additional experiments to better understand the nature of aggregates we detected by PROTEOSTAT. New electron microscopy and confocal microscopy data clearly show that aggresomes are not indeed formed, as we originally presumed. Instead, aggregates appear to coat the nuclei of photoreceptor cells. This is a novel observation and this type of formation of aggregates will need to be characterized further in the future to better understand how aggregates cause photoreceptor cell death. Assignment of aggregate type is not central to our findings, and in the revised manuscript we present the current state of our understanding, which we hope will spur future studies. We respond below to the specific issues related to this concern numbered 1-6.

Reviewer #2:

1) The fluorescence microscopy shows aggregates at too low magnification; they appear to be brightly labeled cell bodies, not subcellular structures. They appear at the level of the outer nuclear layer (i.e. photoreceptor nuclei), which is not consistent with the original description of aggresomes as pericentriolar structures (which would be expected in the inner segment).

Response:

Since assignment of aggregate type was not a primary objective in the original manuscript, we did not conduct more in-depth studies in the original manuscript. As suggested by the reviewer, we have conducted higher magnification confocal microscopy, and additionally collected multiple z-stack images to obtain 3D information, including the generation of 3D movies (Figs. 8C and 8D). These new data reveal a distinct morphology for the PROTEOSTAT staining that was not apparent from low magnification images. The morphology of PROTEOSTAT staining is not consistent with aggregates forming large inclusions such as aggresomes, but rather, aggregates appear to coat the surface of photoreceptor cell nuclei, a behavior that is also observed with some amyloid-type aggregates. This type of morphology of aggregates observed in our mutant mice is a new finding and we thank the Reviewer for suggesting these studies.

Reviewer #2:

2) Although electron microscopy was carried out by the authors for other purposes (i.e. imaging the organization of ROS disks), there is no attempt by the authors to provide evidence of aggresome-like structures at the EM level. Previous studies with extensive electron microscopy on the P23H knock-in animals also did not report aggresome-like structures. (Sakami et al 2014, Sakami et al 2011), but this was a similar scenario of non-reporting, not negative findings.

Response:

We have conducted additional electron microscopy studies to determine whether or not aggresomes are present in the retina of mutant mice. Our new images, which we now present in Supplementary Fig. 3, do not show the presence of large electron dense material that would be indicative of aggresomes. We have revised the manuscript to point out that the misfolding mutants do not appear to form aggresomes, as suggested by the advertised use of PROTEOSTAT, and is also supported by our new confocal microscopy data.

Reviewer #2:

3) In Fig 7, the PROTEOSTAT dye used by the authors strongly labels structures that have not been demonstrated to contain aggresomes (i.e. WT rod outer segments) with no explanation provided by the authors, suggesting that this dye does not exclusively label aggresomes, and/or labels non-aggregated rhodopsin.

Response:

We used the PROTEOSTAT dye to detect rhodopsin aggregates *in vivo*. We focused on staining in the outer nuclear layer, which was distinct from staining in other regions of the retina, including the outer segment, that were fainter and variable in intensity and may represent non-specific staining. We now indicate this non-specific nature of staining in the revised manuscript.

We focused on the distinct staining in the outer nuclear layer since we conducted control studies to ensure this represents aggregates (Fig. 8A). In rhodopsin knockout mice, where photoreceptor cell death occurs but there is no rhodopsin expressed, we do not see PROTEOSTAT staining in the outer nuclear layer. Likewise, in *Prph2^{Rd2}* mice, where rhodopsin is mislocalized and photoreceptor cell death occurs, no PROTEOSTAT staining is evident in the outer nuclear layer. These control studies demonstrate that neither photoreceptor cell death nor mislocalization of rhodopsin results in a PROTEOSTAT-positive signal in the outer nuclear layer, and that PROTEOSTAT staining in the outer nuclear layer of mutant mice expressing P23H or G188R rhodopsin represents aggregates.

Reviewer #2:

4) Similarly, ubiquitin labeling is unexpectedly (to me?) present in OS of WT animals, without explanation.

Response:

We believe that the anti-ubiquitin staining in the outer segments in the original manuscript may be non-specific. The original anti-ubiquitin antibody used was the same one used in the study characterizing PROTEOSTAT as a dye to label aggresomes¹, which showed colocalization of ubiquitin and PROTEOSTAT. This antibody was discontinued, so we were unable to repeat these studies. We repeated experiments with an anti-ubiquitin antibody from Santa Cruz in the revised manuscript. This antibody does not show ubiquitin labeling in the outer segments, indicating that staining with the original antibody may be non-specific. We tested 2 other anti-ubiquitin antibodies, which also showed similar staining patterns as the Santa Cruz antibody.

Reviewer #2:

5) Co-localization with ubiquitin and vimentin labeling with PROTEOSTAT labeling is used as evidence for the presence of aggresomes. However, ubiquitin and vimentin labeling is uniformly brighter in all retinal cell types in knock-in animals of both types, not just rod photoreceptors as would be expected, with no explanation provided by the authors.

Response:

The ubiquitin-proteasome system is found throughout the retina and it is upregulated by stresses imparted on the retina as occurs in retinal degenerative disease^{5,6}. Thus, this increased ubiquitin staining could be the result of this response to stress because of retinal degeneration. As indicated above, we needed to use a new anti-ubiquitin antibody since the original antibody was discontinued. The new anti-ubiquitin antibody does not show a robust increase in signal throughout the retina as the original antibody did, but still shows staining in the outer nuclear layer. We obtained similar results with 2 other anti-ubiquitin antibodies we tested. It is unclear the reason for differences in staining between the original and new anti-ubiquitin antibodies. Ubiquitination is a complex process, and the two antibodies may have different selectivity of different types of ubiquitin. Alternatively, the older antibody may have more non-specific staining. The purpose of our studies was to characterize aggregates and not ubiquitination, so we did not pursue the apparent difference in ubiquitin detection in other regions of the retina. Our new anti-ubiquitin staining demonstrates some colocalization between ubiquitin and

PROTEOSTAT, as was observed in the original manuscript. High-magnification confocal microscopy, however, demonstrates that this colocalization is not associated with DAPI-stained nuclei and is not indicative of aggresomes. These new data are presented in Fig. 8B of the revised manuscript.

Vimentin levels also increase in retinal degeneration due partially to the activation of Müller glial cell ^{7,8}. Müller cell processes are primarily in the inner retina with only a few processes making it to the outer nuclear layer under normal conditions. However, more processes enter the outer nuclear layer when photoreceptor cell death occurs. Staining by the original anti-vimentin antibody, which was used previously to study aggresomes ^{2,3}, appears to reflect these events. The staining pattern, however, did not distinctly show the Müller cell processes, and we are unclear the reason. We have subsequently tested another antibody that does exhibit the distinct morphology of Müller cell processes that extend throughout the retina. We conducted high magnification confocal microscopy to determine the relationship between vimentin and PROTEOSTAT staining, which showed that vimentin surrounds or is adjacent to PROTEOSTAT-stained cells but not in a manner expected for aggresomes. The vimentin staining we observed was indicative of Müller cell processes surrounding these cells. This along with our new high magnification confocal microscopy and EM studies indicate that the aggregates detected by PROTEOSTAT are not indeed aggresomes. We have therefore decided to remove the vimentin data in the revised manuscript.

Reviewer #2:

6) Previous reports indicate vimentin is only expressed in a subset of retinal cells, which does not include photoreceptors. Generally, vimentin is not found in mature neurons; instead, neurofilaments are present.

Response:

As explained above, vimentin will stain Müller glial cells in the retina. It is unclear why our original antibody did not display the distinct morphology of Müller cells. Since our updated data now point to the absence of aggresomes, the studies with vimentin are now moot.

Reviewer #2:

The author is a noted expert on atomic force microscopy of photoreceptors and FRET spectroscopy, with several high-profile publications. These aspects of the work seem well done and are likely high quality, though I am not an expert on either technique. The more standard forms of analysis also seem well done, with the exceptions noted above. In my opinion, the article cannot claim to have identified aggregates/aggresomes in vivo without improving the fluorescence study, providing electron microscopic evidence, and providing western blots from P23H/P23H and G188R/G188R retinas. Currently, the only evidence of in-vivo aggregation is from dye labeling, which also labels outer segments of WT retina containing (presumably) un-aggregated rhodopsin. It is not critical that the mutant aggregated rhodopsin is found in a structure that can be termed an “aggresome”. If the structures differ from canonical aggresomes in notable ways, and have not been previously reported, that would also be interesting.

Response:

We have conducted higher magnification confocal microscopy, including 3D reconstructions, and electron microscopy, as indicated above. These new studies now demonstrate that aggregates do not form aggresomes. Rather, aggregates appear to coat the surface of the nuclei of photoreceptor cells. This is a notable finding on the organization and structure of aggregates of misfolding mutants of rhodopsin. Additionally, we have new Western blot evidence that rhodopsin mutants form aggregates *in vivo*, as described below.

Reviewer #2:

Quote from the abstract: “A distinguishing feature of these mutants *in vitro* is that they mislocalize and aggregate. It is unclear whether these features occur *in vivo* and play a prominent role in causing retinal degeneration.”

This is misleading. Although it has not, to my knowledge, been demonstrated in humans, multiple studies using multiple different animal models of P23H rhodopsin have demonstrated that P23H rhodopsin mislocalizes. The authors note later in the manuscript that the initial study of the P23H knock-in mouse did not identify mis-localization; however, other examples have been reported in other transgenic animals.

Response:

We did not intend to be misleading with this sentence, and as the reviewer notes, mislocalization has been detected in other animal models studied, which we cited in the original manuscript. We have revised this sentence to now read, “It has been questioned whether these features occur *in vivo* and it is unclear whether they play a prominent role in causing retinal degeneration.”

Reviewer #2:

Fig 5B (western blot): there is no evidence of aggregation in this western blot. Is it possible to determine what fraction of the protein is P23H rhodopsin? Much of the rhodopsin signal in this blot is likely WT rhodopsin from OS. What would a blot of P23H/P23H or G188R/G188R retina look like? In this scenario, any aggregation should be much more obvious.

Response:

In both B6 and heterozygous mutant mice, Western blots show that a band corresponding to a rhodopsin monomer is the predominant species with bands corresponding to rhodopsin oligomers also present but at lower intensities. This suggests that the rhodopsin in heterozygous mutant mice largely exhibit wild-type rhodopsin behavior. It has been shown previously that the mutant P23H rhodopsin is largely degraded in heterozygous mutant mice, and we also indicated in the original manuscript that the photoreceptor is tuned to degrade misfolding mutants of rhodopsin. We have taken a more quantitative approach to our original Western blots and now show in the revised manuscript that both heterozygous mutant mice express roughly half the amount of rhodopsin protein as that in B6 mice (Fig. 5D), which is consistent with the notion that most of the misfolding mutants is degraded and that the bands in Western blots correspond mostly to wild-type rhodopsin. We also conducted Western blots on samples from homozygous mutant mice, where only the mutants will be present. There are two major conclusions from this

new blot. First, rhodopsin migrates predominantly not as monomers but larger complexes. This pattern is the same as that observed previously in heterologously expressed misfolding rhodopsin mutants where we detected aggregates by FRET⁴. Thus, this supports the idea that both misfolding mutants of rhodopsin aggregate *in vivo*. Second, the amount of rhodopsin in homozygous mutant mice is only about 2% of that in B6 mice, which supports the idea that most of the mutant is degraded, and that in heterozygous mutant mice, most of the rhodopsin detected in Western blots is in the wild-type form.

Reviewer #2:

Fig 6: No AFM images of WT disks are included. Although this has been previously published, this control would be useful to include to allow the reader to conduct a comparison. Are aggresomes visible in the authors' electron microscopy images? The original descriptions of aggresomes (Johnston and Kopito) included electron microscopy images showing a dense pericentriolar structure surrounded by a cage of filaments. The authors provide no investigation of the inner segments or cell bodies to determine the nature of the structures labeled in the fluorescence study.

Response:

We had not included AFM images of ROS discs from B6 mice since we have published these images several times. We now include a couple examples of AFM images from B6 mice so that the reader will have a reference. Visually, it is difficult to discern differences from AFM images alone, and therefore the more important part of the figure is our quantitative analysis.

As indicated in earlier responses, we have performed new electron microscopy experiments to examine the nature of aggregates observed by PROTEOSTAT staining. These new data do not show any evidence of aggresomes and are presented in the Supplementary Fig. 3

Reviewer #2:

Fig 7C: please comment on the fact that the PROTEOSTAT dye and anti-ubiquitin seem to unexpectedly intensely label rod outer segments in WT mice; what is the evidence that PROTEOSTAT specifically labels aggregated rhodopsin as opposed to all rhodopsin? Please comment on the much higher levels of ubiquitin and vimentin staining in all retinal layers and cell types in the mutant mouse retina. Please comment on the fact that vimentin has been reported to be expressed only in developing neurons, whereas mature neurons express neurofilaments. In the retina, vimentin has been reported to be expressed in Muller cells, astrocytes, and horizontal cells. However, the labeling of WT retina with anti-vimentin in Figure 7C does not appear to be restricted to specific cell types. What is the evidence that the labeled structures are aggresomes and not (e.g.) autophagosomes? A possibility is that the labeled cells are undergoing a proteostatic crisis in which the labeled structures are autophagosomes. This would be consistent with previous publications (Yao and Zacks 2018, Yao and Zacks 2016, Bogea et al 2015, Wen et al., 2019)

Response:

As we indicated in more detail above, we believe the PROTEOSTAT staining in the rod outer segment represents non-specific staining, that the higher levels of ubiquitin and vimentin in mutant mice are a result of stress imparted on the entire retina because of retinal degeneration, and that vimentin staining in the outer nuclear layer likely represents Müller cell processes rather than the structure of aggresomes. We believe that PROTEOSTAT staining in the outer nuclear layer represents aggregated rhodopsin since our control study examining *Prph2^{Rd2}* mice, where rhodopsin is mislocalized to the outer nuclear layer, no PROTEOSTAT staining is observed (Fig. 8A). Our new confocal microscopy and electron microscopy data indicate that aggregates do not form aggresomes, but rather coat the nuclei of photoreceptor cells. The labeled structures do not appear to be autophagosomes.

Reviewer #2:

Figure 8: Why are no results presented for WT mice?

Response:

We did not show results for B6 mice in Fig. 8 of the original manuscript because there was no TUNEL staining or PROTEOSTAT staining in the retina of these mice. This data was moved to Fig. 7 in the revised manuscript, and it is stated in the text that no TUNEL or PROTEOSTAT staining is observed at any aged tested. The value would be 0 in the graphs, and we did think it was necessary to include.

Reviewer #2:

Figure 8 C, D, and discussion: I suggest including more discussion regarding how this model differs significantly from that of Arshavsky and Lobanova, in which it is proposed that retinal degeneration in P23H rhodopsin photoreceptors and other forms of retinal degeneration occurs when the machinery for dealing with misfolded proteins (the proteasome) becomes saturated (“proteostatic crisis”)? (Lobanova et al 2013 and other papers by Lobanova/Arshavsky). Are there novel predictions of this model?

Response:

We have revised the Discussion to provide more mechanistic insight into photoreceptor cell death based on our longitudinal studies, which we did not include in the original manuscript. Figs. 8C and 8D in the original manuscript have been modified accordingly and have been moved to Figs. 9A and 9B in the revised manuscript. We observe an exponential loss of photoreceptor cells in both mutant mouse models (Figs. 4E and 4F) and both models exhibit an early peak in cell death followed by decreasing cell death (Figs. 7b and 7C), which indicates that cell death occurs stochastically rather than by mechanisms related to the cumulative damage hypothesis⁹. In the cumulative damage hypothesis, photoreceptor cell death occurs because of a gradual accumulation of toxic species that occurs with age, which would result in increased risk of cell death with age and sigmoidal kinetics of photoreceptor cell death, neither of which is observed in our longitudinal studies. Since aggregation mirrors cell death, it is not the gradual accumulation of aggregates over time but rather a stochastic event that results in appreciable aggregation that appears to cause photoreceptor cell death. The implication of a stochastic

mechanism is that remaining photoreceptor cells should be functional, which we observe here (Figs. 4G and 4H), and that therapeutic intervention can occur at any time as long as photoreceptor cells are present. This mechanism is critical for a proper understanding of photoreceptor cell death in these instances of adRP and in developing therapeutic strategies.

We cannot say currently whether or not our presented mechanism is in contradiction to that suggested by Arshavsky and Lobanova. Longitudinal studies were not conducted in their studies, so it is difficult to assess the correlation between proteasome insufficiency and photoreceptor cell death. In fact, kinetic studies are currently limited but required to better understand mechanism. It is quite possible that proteosomal insufficiency contributes to the aggregation that we observe, making our study consistent with their observations. But it is too premature to make such a conclusion. In the revised Discussion, we now indicate that the trigger for stochastic aggregation may involve failure in some aspect of the proteostasis network including the chaperone and a quality control system, proteasome activity or autophagy.

Reviewer #2:

Quote: “Smaller aggregates of misfolded rhodopsin mutants are also likely present in the retina of the mutant mice investigated here as well, although we are currently unable gain access to this information.” This is unusual phrasing. What does this mean? Techniques do not exist, or are not accessible to the authors, or that the information already exists somewhere that is inaccessible to the authors?

Response:

The intended meaning was that techniques were not available to detect smaller aggregates *in vivo*. In the revised manuscript, this sentence was removed as it was no longer necessary based on the new data and revised text.

References

- 1 Shen, D. *et al.* Novel cell- and tissue-based assays for detecting misfolded and aggregated protein accumulation within aggresomes and inclusion bodies. *Cell Biochem. Biophys.* **60**, 173-185 (2011). <https://doi.org:10.1007/s12013-010-9138-4>
- 2 Johnston, J. A., Ward, C. L. & Kopito, R. R. Aggresomes: a cellular response to misfolded proteins. *J. Cell Biol.* **143**, 1883-1898 (1998). <https://doi.org:10.1083/jcb.143.7.1883>
- 3 Tiwari, A. *et al.* Caveolin-1 is an aggresome-inducing protein. *Sci Rep* **6**, 38681 (2016). <https://doi.org:10.1038/srep38681>
- 4 Gragg, M., Kim, T. G., Howell, S. & Park, P. S. Wild-type opsin does not aggregate with a misfolded opsin mutant. *Biochim. Biophys. Acta* **1858**, 1850-1859 (2016). <https://doi.org:10.1016/j.bbame.2016.04.013>
- 5 Plafker, S. M. Oxidative stress and the ubiquitin proteolytic system in age-related macular degeneration. *Adv. Exp. Med. Biol.* **664**, 447-456 (2010). https://doi.org:10.1007/978-1-4419-1399-9_51

- 6 Esteve-Rudd, J., Campello, L., Herrero, M. T., Cuenca, N. & Martin-Nieto, J. Expression in the mammalian retina of parkin and UCH-L1, two components of the ubiquitin-proteasome system. *Brain Res.* **1352**, 70-82 (2010).
<https://doi.org/10.1016/j.brainres.2010.07.019>
- 7 Fernandez-Sanchez, L., Lax, P., Campello, L., Pinilla, I. & Cuenca, N. Astrocytes and Muller Cell Alterations During Retinal Degeneration in a Transgenic Rat Model of Retinitis Pigmentosa. *Front Cell Neurosci* **9**, 484 (2015).
<https://doi.org/10.3389/fncel.2015.00484>
- 8 Hippert, C. *et al.* Muller glia activation in response to inherited retinal degeneration is highly varied and disease-specific. *PLoS ONE* **10**, e0120415 (2015).
<https://doi.org/10.1371/journal.pone.0120415>
- 9 Clarke, G. *et al.* A one-hit model of cell death in inherited neuronal degenerations. *Nature* **406**, 195-199 (2000). <https://doi.org/10.1038/35018098>

REVIEWERS' COMMENTS

Reviewer #1 (Remarks to the Author):

Revision comments

The authors have addressed many of my comments; however, I think there are a few minor revisions required in the MS to clarify their findings.

Major comment 1 and 2.

The higher magnification proteostat images and clarification that these are not 'aggresomes', as well as the absence of electron dense 'aggregates' in the EM is welcome and the MS is improved as a result. However, it does place a greater demand on an orthogonal method to show 'aggregation' in vivo. The additional western blot data in 5B-D are not compelling evidence for aggregation. Dimer and higher oligomers are also visible in the B6 samples, where there is no aggregation, with no pronounced increase in the dimer or higher molecular weight species. The major difference is the reduction in the monomer band.

I think it would be better if the authors discussed in more detail what the proteostat might be detecting and what sort of species these were as opposed to using these as evidence of aggregation. It would also be better if the abstract described the proteostat staining as opposed to stating it is 'aggregation'.

Major comment 3.

I think the evidence here that OS effects are not caused by G188R reaching the OS is indirect. Whilst their hypothesis might be correct that the OS disruption might be due to photoreceptor distress and be unrelated to the presence of mutant rhodopsin, residual P23H has been shown to reach the OS and lead to disruption of structure, it might be better if the authors discussed the possibility that a small amount of G188R might also reach the OS if it escaped the ER and degradation.

Minor Comment 1.

Unfortunately, I do not think the revised abstract is appropriate. "It has been questioned whether these features occur in vivo and it is unclear whether they play a prominent role in causing retinal degeneration." implies that mislocalization has not been shown in vivo before, which it has, so this needs to be modified. In addition, I am not sure that this study clarifies if mislocalization and aggregation 'cause' retinal degeneration either.

Reviewer #2 (Remarks to the Author):

This manuscript has been substantially revised from the previous submission, and has been significantly improved. The authors have resolved my previously stated concerns. However, I also recommend some minor changes:

- 1) There is significant referral in the manuscript to the morphology of photoreceptor nuclei. However, in the images supplied, the nuclei are rendered in the blue channel of RGB images, which makes it quite difficult to visualize detail; human eyes are poor at resolving detail presented in this way. Some possibilities include: a) render the lower image in each panel of 8D (isolated blue channel) using greyscale, rather than the blue channel (i.e. copy the blue channel info to all three channels of the image, or re-render in greyscale using the microscope software). b) switch the blue channel to the green channel in these images by similar methods c) render the images in red for proteostat staining and greyscale for nuclei via similar methods. Or, perhaps the authors have alternative ideas for improving visualization of nuclear morphology.
- 2) The outer nuclear membrane is continuous with the ER; most likely aggregates are delivered via this continuity. Alternatively is it possible they are in a region of the ER that surrounds the nuclei? Are membranes distinct from the PM and nuclear envelope visible in EM micrographs such as supplemental figure 3?
- 3) An interesting question is whether the aggregation around nuclei observed with proteostat staining causes cell death, or is a symptom of cell death. I do not think this is resolvable using this data set. However, it would be interesting to discuss the following point: a) in images such as those in Figure 8 B, C, and D, and supplemental Figure 2, there are cells with both abnormal nuclear morphology and proteostat staining. The text states that not all cells stained with proteostat have abnormal nuclear morphologies. However, do all cells with abnormal nuclear morphologies stain with proteostat? If yes, it suggests that accumulation of aggregates at least represents an early step in cell death.
- 4) The possibility that cells go through a sudden appearance of aggregated protein before undergoing a stochastic cell death process is similar to the one-hit hypothesis of Clarke et al (already cited by the authors) and also late-career papers by Max Perutz that dealt with mechanisms underlying Huntington's disease, suggesting that nucleation of aggregates initiates cell death (Nature 2001).

Manuscript NCOMMS-23-15269A

“Aggregation of rhodopsin mutants in mouse models of autosomal dominant retinitis pigmentosa”

We thank both reviewers for their valuable comments and suggestions on both our original and revised manuscripts. The changes in response to the reviewers’ comments have significantly improved the manuscript, as was acknowledged by each reviewer. We have made additional changes in response to the comments below and believe the changes have strengthened the manuscript further. Provided below are our responses to specific comments made by each reviewer. Please note that the page numbers indicated in the response refer to those in the manuscript file with tracked changes highlighted.

Response to Reviewer #1

Reviewer #1:

The authors have addressed many of my comments; however, I think there are a few minor revisions required in the MS to clarify their findings.

Response:

We thank the Reviewer for all their useful comments that have helped to improve our manuscript. We have also addressed the new comments below, which we hope has further clarified the outcome and conclusions of our findings.

Reviewer #1:

Major comment 1 and 2.

The higher magnification proteostat images and clarification that these are not ‘aggresomes’, as well as the absence of electron dense ‘aggregates’ in the EM is welcome and the MS is improved as a result. However, it does place a greater demand on an orthogonal method to show ‘aggregation’ *in vivo*. The additional western blot data in 5B-D are not compelling evidence for aggregation. Dimer and higher oligomers are also visible in the B6 samples, where there is no aggregation, with no pronounced increase in the dimer of higher molecular weight species. The major difference is the reduction in the monomer band.

Response:

We thank the reviewer for their suggestions to use higher magnification confocal microscopy and electron microscopy to characterize the apparent aggregates of rhodopsin mutants. We are happy to see our new data has improved our manuscript.

We agree with the reviewer that the appearance of bands in Western blots corresponding to higher order molecular weight species in and of itself does not provide evidence for *in vivo* aggregation or complex formation. However, we disagree that the Western blots we present do not provide evidence for the aggregation of mutant rhodopsin from mice. We demonstrated in our previous work that the G188R rhodopsin mutant expressed in HEK293 cells and shown to aggregate by the same FRET procedure used in the current study, displays a distinct banding

pattern on Western blots where the band corresponding to a monomer is absent and only higher order species are detected (Gragg et al. (2016) *Biochim. Biophys. Acta* 1858, 1850-1859). This is the same banding pattern observed in Western blots presented here for homozygous mutant mice (Fig. 5C). This suggests that the mutants in mice share the same propensity to aggregate as those expressed in HEK239 cells, where we explicitly tested for aggregation. We have clarified this in the Discussion on Page 21. As suggested by the Reviewer, we removed the part of sentence in the Results section on Page 11 that reads, "...which is indicative of rhodopsin aggregation." We decided to remove this sentence since the details we write in the Discussion section are required to understand why the banding pattern on the Western blot is suggestive of aggregation.

Reviewer #1:

I think it would be better if the authors discussed in more detail what the proteostat might be detecting and what sort of species these were as opposed to using these as evidence of aggregation. It would also be better if the abstract described the proteostat staining as opposed to stating it is 'aggregation'.

Response:

Our intent was not to equate PROTEOSTAT staining directly to rhodopsin mutant aggregation in the absence of other observations. We believe our control studies in Fig. 8A are part of the discussion requested by the reviewer on the species that is being detected by PROTEOSTAT. Control studies in *Rho*^{-/-} mice demonstrate that PROTEOSTAT staining in the outer nuclear layer is not detecting a non-rhodopsin species generated as a byproduct of retinal degeneration. Control studies in *Prph2*^{Rd2} mice demonstrate that retinal degeneration does not cause mislocalized WT rhodopsin to form a species, such as an aggregate, that can be stained by PROTEOSTAT. These controls rule out alternate explanations for PROTEOSTAT staining in the outer nuclear layer. Unless there are alternate explanations we have not considered, the only other explanation appears to be that PROTEOSTAT is staining rhodopsin mutant aggregates. These controls along with our *in vitro* data and Western blot data in Fig 5C provide evidence that rhodopsin mutants appear to form aggregates in the mutant mice. We have revised the manuscript to make this clearer in a paragraph in the Discussion on Pages 21.

We have tried to make changes throughout the manuscript where we indicate that PROTEOSTAT is direct evidence for rhodopsin aggregation. We have tried to be more careful in our writing to indicate that what we observe is PROTEOSTAT staining rather than aggregation. We have made changes to the Abstract to better reflect this dichotomy.

Reviewer #1:

Major comment 3.

I think the evidence here that OS effects are not caused by G188R reaching the OS is indirect. Whilst their hypothesis might be correct that the OS disruption might be due to photoreceptor distress and be unrelated to the presence of mutant rhodopsin, residual P23H has been shown to reach the OS and lead to disruption of structure, it might be better if the authors discussed the possibility that a small amount of G188R might also reach the OS if it escaped the ER and degradation.

Response:

We believe that the more severe aggregation profile of the G188R rhodopsin mutant makes it unlikely it would reach the outer segment to the extent observed for the P23H rhodopsin mutant. We agree with the reviewer that we did not explicitly test this assumption and therefore cannot completely rule out the possibility. We have decided to remove the sentence on Page 23 indicating that the disruption of outer segment disc morphogenesis in mutant mice is caused by retinal degeneration and left the question of why this process is disrupted open-ended. We have added a sentence on Page 24 indicating that the question of whether the G188R rhodopsin mutant reaches the outer segment must be tested explicitly in the future.

Reviewer #1:**Minor Comment 1.**

Unfortunately, I do not think the revised abstract is appropriate. “It has been questioned whether these features occur *in vivo* and it is unclear whether they play a prominent role in causing retinal degeneration.” implies that mislocalization has not been shown *in vivo* before, which it has, so this needs to be modified. In addition, I not sure that this study clarifies if mislocalization and aggregation ‘cause’ retinal degeneration either.

Response:

We have modified the sentence, “It has been questioned whether these features occur *in vivo* and it is unclear whether they play a prominent role in causing retinal degeneration” to now read, “It is unclear whether or not these features contribute to retinal degeneration *in vivo*.” We have also revised the last sentence of the Abstract to indicate that our study indicates a “potential” rather than “prominent” role for rhodopsin aggregation in retinal degeneration.

Response to Reviewer #2**Reviewer #2:**

This manuscript has been substantially revised from the previous submission, and has been significantly improved. The authors have resolved my previously stated concerns. However, I also recommend some minor changes:

Response:

We thank the reviewer for providing constructive comments and suggestions, which we believe has significantly improved our manuscript. We are happy to see that the reviewer also agrees with this sentiment.

Reviewer #2:

1) There is significant referral in the manuscript to the morphology of photoreceptor nuclei. However, in the images supplied, the nuclei are rendered in the blue channel of RGB images, which makes it quite difficult to visualize detail; human eyes are poor at resolving detail presented in this way. Some possibilities include: a) render the lower image in each panel of 8D

(isolated blue channel) using greyscale, rather than the blue channel (i.e. copy the blue channel info to all three channels of the image, or re-render in greyscale using the microscope software). b) switch the blue channel to the green channel in these images by similar methods c) render the images in red for proteostat staining and greyscale for nuclei via similar methods. Or, perhaps the authors have alternative ideas for improving visualization of nuclear morphology.

Response:

We have taken the Reviewer's advice and have changed the blue coloring to greyscale for the images of nuclei in Fig. 8D. As the reviewer predicted, the greyscale coloring better displays healthy nuclei with a single chromocenter versus unhealthy nuclei with condensed chromatin. We have also grouped and ordered the nuclei in Fig. 8D accordingly to whether they appear healthy or unhealthy and describe this in the legend to the figure. We have also introduced 3D surface rendered images, which are described below, to better discern the morphology of DAPI and PROTEOSTAT staining.

Reviewer #2:

2) The outer nuclear membrane is continuous with the ER; most likely aggregates are delivered via this continuity. Alternatively is it possible they are in a region of the ER that surrounds the nuclei? Are membranes distinct from the PM and nuclear envelope visible in EM micrographs such as supplemental figure 3?

Response:

We have added a 3D surface rendering representation of PROTEOSTAT staining in Fig. 8E to better illustrate that PROTEOSTAT staining surrounds the nucleus. More work needs to be done to define this localization more precisely. We had done some colocalization studies of PROTEOSTAT with calnexin, which did not reveal any colocalization. This indicated that the PROTEOSTAT stained species is not in the endoplasmic reticulum, but we cannot rule out the involvement of endoplasmic reticulum membrane directly associated with the nuclear envelope. The PROTEOSTAT staining does not appear to be associated with the plasma membrane. We do observe the nuclear envelope in EM micrographs, and they may appear darker for some nuclei of mutant mice. We were not confident in these observations without further investigation and therefore did not indicate this in the manuscript.

Reviewer #2:

3) An interesting question is whether the aggregation around nuclei observed with proteostat staining causes cell death, or is a symptom of cell death. I do not think this is resolvable using this data set. However, it would be interesting to discuss the following point: a) in images such as those in Figure 8 B, C, and D, and supplemental Figure 2, there are cells with both abnormal nuclear morphology and proteostat staining. The text states that not all cells stained with proteostat have abnormal nuclear morphologies. However, do all cells with abnormal nuclear morphologies stain with proteostat? If yes, it suggests that accumulation of aggregates at least represents an early step in cell death.

Response:

We agree with the reviewer that this is an important question and must be addressed moving forward. Our assessment of the health of nuclei was purely based on our visual assessment of nuclear morphology. The observation that visually healthy nuclei could be coated by PROTEOSTAT staining certainly leaves open the possibility that this process precedes cell death. We do see some unhealthy nuclei with condensed chromatin without PROTEOSTAT staining, however, we are unsure if this is just an artifact of incomplete staining or sectioning. More work needs to be done using proper markers of different time points of cell death to address this important question. We indicate in the revised manuscript on Page 22 a short sentence discussing this issue.

Reviewer #2:

4) The possibility that cells go through a sudden appearance of aggregated protein before undergoing a stochastic cell death process is similar to the one-hit hypothesis of Clarke et al (already cited by the authors) and also late-career papers by Max Perutz that dealt with mechanisms underlying Huntington's disease, suggesting that nucleation of aggregates initiates cell death (Nature 2001).

Response:

We thank the Reviewer for bringing to our attention this Hypothesis piece written by Max Perutz. We have incorporated this paper into our Discussion on Page 25.